# Learning Manifold Dimensions with Conditional Variational Autoencoders

**Yijia Zheng**[1]* **Tong He**[2] **Yixuan Qiu**[3] **David Wipf**[2]
[1] Department of Statistics, Purdue University
[2] Amazon Web Services
[3] School of Statistics and Management, Shanghai University of Finance and Economics
zheng709@purdue.edu, {htong, daviwipf}@amazon.com, qiuyixuan@sufe.edu.cn

## Abstract

Although the variational autoencoder (VAE) and its conditional extension (CVAE) are capable of state-of-the-art results across multiple domains, their precise behavior is still not fully understood, particularly in the context of data (like images) that lie on or near a low-dimensional manifold. For example, while prior work has suggested that the globally optimal VAE solution can learn the correct manifold dimension, a necessary (but not sufficient) condition for producing samples from the true data distribution, this has never been rigorously proven. Moreover, it remains unclear how such considerations would change when various types of conditioning variables are introduced, or when the data support is extended to a union of manifolds (e.g., as is likely the case for MNIST digits and related). In this work, we address these points by first proving that VAE global minima are indeed capable of recovering the correct manifold dimension. We then extend this result to more general CVAEs, demonstrating practical scenarios whereby the conditioning variables allow the model to adaptively learn manifolds of varying dimension across samples. Our analyses, which have practical implications for various CVAE design choices, are also supported by numerical results on both synthetic and real-world datasets.

## 1 Introduction

Variational autoencoders (VAE) [6, 14] and conditional variants (CVAE) [17] are powerful generative models that produce competitive results in various domains such as image synthesis [5, 13, 21], natural language processing [16], time-series forecasting [9, 19], and trajectory prediction [8]. As a representative example, when equipped with an appropriate deep architecture, VAE models have recently achieved state-of-the-art performance generating large-scale images [11]. And yet despite this success, there remain VAE/CVAE behaviors in certain regimes of interest where we lack a precise understanding or a supporting theoretical foundation.

In particular, when the data lie on or near a low-dimensional manifold, as occurs with real-world images [12], it is meaningful to have a model that learns the manifold dimension correctly. The latter can provide insight into core properties of the data and be viewed as a necessary, albeit not sufficient, condition for producing samples from the true distribution. Although it has been suggested in prior work [3, 4] that a VAE model can learn the correct manifold dimension when globally optimized, this has only been formally established under the assumption that the decoder is linear or affine [2]. And the potential ability to learn the correct manifold dimension becomes even more nuanced when conditioning variables are introduced. In this regard, a set of discrete conditions (e.g., MNIST image digit labels) may correspond with different "slices" through the data space, with each

---

*Work completed during internship at the AWS Shanghai AI Labs.

36th Conference on Neural Information Processing Systems (NeurIPS 2022).

inducing a manifold with varying dimension (intuitively, the manifold dimension of images labelled "1" is likely smaller than those of "5"). Alternatively, it is possible to have data expand fully in the ambient space but lie on a low-dimensional manifold when continuous conditional variables are present. Such a situation can be trivially constructed by simply treating some data dimensions, or transformations thereof, as the conditioning variables. In both scenarios, the role of CVAE models remains under-explored.

Moreover, unresolved CVAE properties in the face of low-dimensional data structure extend to practical design decisions as well. For example, there has been ongoing investigation into the choice between a fixed VAE decoder variance and a learnable one [3, 4, 10, 15, 18], an issue of heightened significance when conditioning variables are involved. And there exists similar ambiguity regarding the commonly-adopted strategy of sharing weights between the prior and encoder/posterior in CVAEs [7, 17]. Although perhaps not obvious at first glance, in both cases these considerations are inextricably linked to the capability of learning data manifold dimensions.

Against this backdrop our paper makes the following contributions:

(i) In Section 2.1 we provide the first demonstration of general conditions under which VAE global minimizers provably learn the correct data manifold dimension.

(ii) We then extend the above result in Section 2.2 to address certain classes of CVAE models with either continuous or discrete conditioning variables, the latter being associated with data lying on a union of manifolds.

(iii) Later, Section 3 investigates common CVAE model designs and training practices, including the impact of strategies for handling the decoder variance as well as the impact of weight sharing between conditional prior and posterior networks.

(iv) Section 4 supports our theoretical conclusions and analysis with numerical experiments on both synthetic and real-world datasets.

## 2 Learning the Dimension of Data Manifolds

In this section we begin with analysis that applies to regular VAE models with no conditioning. We then later extend these results to more general CVAE scenarios.

### 2.1 VAE Analysis

We begin with observed variables $x \in \mathcal{X} \subseteq \mathbb{R}^d$, where $\mathcal{X}$ is the ambient data space equipped with some ground-truth probability measure $\omega_{gt}$. Hence the probability mass of an infinitesimal $dx$ on $\mathcal{X}$ is $\omega_{gt}(dx)$ and $\int_{\mathcal{X}} \omega_{gt}(dx) = 1$. VAE models attempt to approximate this measure with a parameterized distribution $p_\theta(x)$ instantiated as marginalization over latent variables $z \in \mathbb{R}^\kappa$ as in $p_\theta(x) = \int p_\theta(x|z)p(z)dz$. Here $p(z) = N(z|0, I)$ is a standardized Gaussian prior and $p_\theta(x|z)$ represents a parameterized likelihood function that is typically referred to as the decoder.

To estimate decoder parameters $\theta$, the canonical VAE training loss is formed as a bound on the average negative log-likelihood given by

$$\mathcal{L}(\theta, \phi) = \int_{\mathcal{X}} \{-\mathbb{E}_{q_\phi(z|x)}[\log p_\theta(x|z)] + \mathbb{KL}[q_\phi(z|x)||p(z)]\}\omega_{gt}(dx) \geq -\int_{\mathcal{X}} \log p_\theta(x)\omega_{gt}(dx),$$

(1)

where the latent posterior distribution $q_\phi(z|x)$ (or the VAE encoder) controls the tightness of the bound via trainable parameters $\phi$. Borrowing from [4], we package widely-adopted VAE modeling assumptions into the following definition:

**Definition 1 ($\kappa$-simple VAE)** *A $\kappa$-simple VAE is a VAE model with dim[z] = $\kappa$ latent dimensions, the Gaussian encoder $q_\phi(z|x) = N(z|\mu_z(x; \phi), \text{diag}\{\sigma_z^2(x; \phi)\})$, the Gaussian decoder $p_\theta(x|z) = N(x|\mu_x(z; \theta), \gamma I)$, and the prior $p(z) = N(z|0, I)$. Here $\gamma > 0$ is a trainable scalar included within $\theta$, while the mean functions $\mu_z(x; \phi)$ and $\mu_x(z; \theta)$ are arbitrarily-complex L-Lipschitz continuous functions; the variance function $\sigma_z^2(x; \phi)$ can be arbitrarily complex with no further constraint.*

Our goal in this section will be to closely analyze the behavior of $\kappa$-simple VAE models when trained on data restricted to low-dimensional manifolds defined as follows:

**Definition 2 (Data lying on a manifold)** *Let $r$ and $d$ denote two positive integers with $r < d$. Then $\mathcal{M}_r$ is a simple $r$-Riemannian manifold embedded in $\mathbb{R}^d$ when there exists a diffeomorphism $\varphi$ between $\mathcal{M}_r$ and $\mathbb{R}^r$. Specifically, for every $x \in \mathcal{M}_r$, there exists a $u = \varphi(x) \in \mathbb{R}^r$, where $\varphi$ is invertible and both $\varphi$ and $\varphi^{-1}$ are differentiable.*

As pointed out in [4], when training $\kappa$-simple VAEs on such manifold data, the optimal decoder variance will satisfy $\gamma \to 0$ (i.e, unbounded from below). And as we will soon show, one effect of this phenomena can be to selectively push the encoder variances along certain dimensions of $z$ towards zero as well, ultimately allowing these dimensions to pass sufficient information about $x$ through the latent space such that the decoder can produce reconstructions with arbitrarily small error. To formalize these claims, we require one additional definition:

**Definition 3 (Active VAE latent dimensions)** *Let $\{\theta_\gamma^*, \phi_\gamma^*\}$ denote globally-optimal parameters of a $\kappa$-simple VAE model applied to (1) as a function of an arbitrary fixed $\gamma$. Then a dimension $j \in \{1, \ldots, \kappa\}$ of latent variable $z$ is defined as an active dimension (associated with sample $x$) if the corresponding optimal encoder variance satisfies $\sigma_z(x; \phi_\gamma^*)_j^2 \to 0$ as $\gamma \to 0$.*

We now arrive at the main result for this section:

**Theorem 1 (Learning the data manifold dimension using VAEs)** *Suppose $\mathcal{X} = \mathcal{M}_r$ with $r < d$. Then for all $\kappa \geq r$, any globally-optimal $\kappa$-simple VAE model applied to (1) satisfies the following:*

*(i)* $\mathcal{L}(\theta_\gamma^*, \phi_\gamma^*) = (d - r) \log \gamma + O(1),$

*(ii) The number of active latent dimensions almost surely equals $r$, and*

*(iii) The reconstruction error almost surely satisfies $\mathbb{E}_{q_{\phi_\gamma^*}(z|x)} \left[ ||x - \mu_x(z; \theta_\gamma^*)||^2 \right] = O(\gamma).$*

While all proofs are deferred to the appendices, we provide a high-level sketch here. First, we prove by contradiction that there must exist at least $r$ active dimensions with corresponding encoder variances tending to zero at a rate of $O(\gamma)$. If this is not the case, we show that the reconstruction term will grow at a rate of $O(\frac{1}{\gamma})$, leading to an overall loss that is unbounded from above. Next, we obtain upper bound and lower bounds on (1), both of which scale as $(d - r) \log \gamma + O(1)$ when the number of active dimensions is $r$. And lastly, we pin down the exact number of active dimensions by showing that the inclusion of any unnecessary active dimensions decreases the coefficient of the $\log \gamma$ scale factor, i.e., the factor $(d - r)$ uniquely achieves the minimal loss.

Overall, Theorem 1 provides a number of revealing insights into the VAE loss surface under the stated conditions. First, we observe that although the loss can in principle be driven to minus infinity via sub-optimal solutions as $\gamma \to 0$, globally-optimal solutions nonetheless achieve an optimal rate (i.e., largest possible coefficient on the $\log \gamma$ factor) as well as the minimal number of active latent dimensions, which matches the ground-truth manifold dimension $r$. Moreover, this all occurs while maintaining a reconstruction error that tends to zero as desired. Additionally, while we have thus far treated $\gamma$ as a manually controlled parameter tending to zero for analysis purposes, when we transition to a trainable setting, similar intuitions are still applicable. More specifically, around global minimizers, the corresponding optimal $\gamma^*$ scales as $\frac{L^2}{d} ||\sigma_z(x; \theta^*)_{1:r}||^2$, where $\sigma_z(x; \theta^*)_{1:r}$ denotes the $r$ optimal latent posterior standard deviations associated with active dimensions. Hence both the decoder variance $\gamma$ and the latent posterior variances of active dimensions converge to zero in tandem at the same rate; see the proof for more details. In contrast, along the remaining inactive dimensions we show that $\lim_{\gamma \to 0} \sigma_z(x; \phi_\gamma^*) = \Omega(1)$, which optimzes the KL term without compromising the reconstruction accuracy.

In closing this section, we note that previous work [4] has demonstrated that global minima of VAE models can achieve zero reconstruction error for all samples lying on a data manifold. But it was *not* formally established in a general setting that this perfect reconstruction was possible using a *minimal* number of active latent dimensions, and hence, it is conceivable for generated samples involving a larger number of active dimensions to stray from this manifold. In contrast, to achieve perfect reconstruction using the minimal number of active latent dimensions, as we have demonstrated here under the stated assumptions, implies that generated samples must also lie on the manifold. Critically, the noisy signals from inactive dimensions are blocked by the decoder and therefore cannot produce deviations from the manifold.

## 2.2 Extension to Conditional VAEs

In this section, we extend our analysis to include conditional models, progressing from VAEs to CVAEs. For this purpose, we introduce conditioning variables $c$ drawn from some set $\mathcal{C}$ with associated probability measure $\nu_{gt}$ such that $\int_{\mathcal{C}} \nu_{gt}(dc) = 1$. Moreover, for any $c \in \mathcal{C}$ there exists a subset $\mathcal{X}_c \subseteq \mathcal{X}$ with probability measure $\omega_{gt}^c$ satisfying $\int_{\mathcal{X}_c} \omega_{gt}^c(dx) = 1$. Collectively we also have $\int_{\mathcal{C}} \int_{\mathcal{X}_c} \omega_{gt}^c(dx)\nu_{gt}(dc) = 1$. Given these definitions, the canonical CVAE loss is defined as

$$
\begin{aligned}
\mathcal{L}(\theta, \phi) &= \int_{\mathcal{C}} \int_{\mathcal{X}_c} \left\{ -\mathbb{E}_{q_\phi(z|x,c)}[\log p_\theta(x|z,c)] + \mathbb{KL}[q_\phi(z|x,c)||p_\theta(z|c)] \right\} \omega_{gt}^c(dx)\nu_{gt}(dc) \\
&\geq \int_{\mathcal{C}} \int_{\mathcal{X}_c} -\log p_\theta(x|c)\omega_{gt}^c(dx)\nu_{gt}(dc),
\end{aligned}
\tag{2}
$$

which forms an upper bound on the conditional version of the expected negative log-likelihood. We may then naturally extend the definition of the $\kappa$-simple VAE model to the conditional regime as follows:

**Definition 4 ($\kappa$-simple CVAE)** *A $\kappa$-simple CVAE is an extension of the $\kappa$-simple VAE with the revised conditional, parameterized prior $p_\theta(z|c) = N(z|\mu_z(c; \theta), \mathrm{diag}\{\sigma_z^2(c; \theta)\})$, the Gaussian encoder $q_\phi(z|x,c) = N(z|\mu_z(x,c; \phi), \mathrm{diag}\{\sigma_z^2(x,c; \phi)\})$, and the Gaussian decoder $p_\theta(x|z,c) = N(x|\mu_x(z,c; \theta), \gamma I)$. The encoder/decoder mean functions $\mu_z(x,c; \phi)$ and $\mu_x(z,c; \theta)$ are arbitrarily-complex L-Lipschitz continuous functions, while the prior mean $\mu_z(c; \theta)$ and variance $\sigma_z^2(c; \theta)$,[2] and encoder variance $\sigma_z^2(x,c; \phi)$ can all be arbitrarily-complex functions with no further constraint.*

Likewise, we may also generalize the definition of active latent dimensions, where we must explicitly account for the modified conditional prior distribution which controls the relative scaling of the data.

**Definition 5 (Active CVAE latent dimensions)** *Let $\{\theta_\gamma^*, \phi_\gamma^*\}$ denote globally-optimal parameters of a $\kappa$-simple CVAE model applied to (2) as a function of an arbitrary fixed $\gamma$. Then a dimension $j \in \{1, \ldots, \kappa\}$ of latent variable $z$ is defined as an active dimension (associated with sample pair $\{x, c\}$) if the corresponding $j$-th optimal encoder/prior variance ratio satisfies $\sigma_z(x,c; \phi_\gamma^*)_j^2 / \sigma_z(c; \theta_\gamma^*)_j^2 \to 0$ as $\gamma \to 0$.*

Note that a CVAE (or VAE) with a standardized (parameter-free) Gaussian prior, the prior variance equals one. Hence it follows that Definition 3 is a special case of Definition 5 where $\sigma_z^2(c; \theta) = I$.

Before proceeding to our main result for this section, there is one additional nuance to our manifold assumptions underlying conditional data. Specifically, if $c$ follows a continuous distribution, then it can reduce the number of active latent dimensions needed for obtaining perfect reconstructions. To quantify this effect, let $\mathcal{M}_r^c \subseteq \mathcal{M}_r$ denote the subset of $\mathcal{M}_r$ associated with $\mathcal{X}_c$. Intuitively then, depending on the information about $x$ contained in $c$, the number of active latent dimensions within $\mathcal{M}_r^c$ may be less than $r$. We quantify this reduction via the following definition:

**Definition 6 (Effective dimension of a conditioning variable)** *Given an integer $t \in \{0, \ldots, r\}$, let $\mathcal{C}_t$ denote a subset of $\mathcal{C}$ with the following properties: (i) There exists a function $g : \mathcal{C}_t \to \mathbb{R}^t$ as well as $t$ dimensions of $\varphi(x)$ denoted as $\varphi(x)_t$, such that $g(c) = \varphi(x)_t$ for all pairs $\{(c, x) : c \in \mathcal{C}_t, \ x \in \mathcal{M}_r^c\}$, where $\varphi$ is a diffeomorphism per Definition 2; and (ii) there does not exist such a function $g$ for $t + 1$. We refer to $t$ as the effective dimension of any conditioning variable $c \in \mathcal{C}_t$.*

Loosely speaking, Definition 6 indicates that any $c \in \mathcal{C}_t$ can effectively be used to reconstruct $t \leq r$ dimensions of $x$ within $\mathcal{M}_r^c$. Incidentally, if $t = r$, then this definition implies that $x$ degenerates to a deterministic function of $c$. Given these considerations, Theorem 1 can be extended to CVAE models conditioned on continuous variables as follows:

**Theorem 2 (Learning the data manifold dimension using CVAEs)** *Suppose $\mathcal{C} = \mathcal{C}_t$ and $\mathcal{X}_c = \mathcal{M}_r^c$ with $r \geq 1$, $t \geq 0$, and $r \geq t$. Then any globally-optimal $\kappa$-simple CVAE model applied to the loss (2), with $\kappa \geq r$, satisfies the following:*

---

[2]While the parameters of the prior mean and variance functions are labeled as $\theta$, this is merely to follow standard convention and group all parameters from the generative pipeline together under the same heading; it is not meant to imply that the decoder and prior actually share the same parameters.

*(i)* $\mathcal{L}(\theta_\gamma^*, \phi_\gamma^*) = (d - r + t) \log \gamma + O(1)$*, and*

*(ii) The number of active latent dimensions almost surely equals $r - t$, and*

*(iii) The reconstruction error almost surely satisfies $\mathbb{E}_{q_{\phi_\gamma^*}(z|x,c)} \left[ ||x - \mu_x(z, c; \theta_\gamma^*)||^2 \right] = O(\gamma)$.*

This result indicates that conditioning variables can further reduce the CVAE loss (by increasing the coefficient on the $\log \gamma$ term as $\gamma \to 0$ around optimal solutions. Moreover, conditioning can replace active latent dimensions; intuitively this occurs because using $c$ to reconstruct dimensions of $x$, unlike dimensions of $z$, incurs no cost via the KL penalty term. Additionally, it is worth mentioning that even if the observed data itself is not strictly on a manifold (meaning $r = d$), once the conditioning variables $c$ are introduced, manifold structure can be induced on $\mathcal{X}_c$, i.e., with $t > 0$ it follows that $d - r + t > 0$ and the number of active latent dimensions satisfies $r - t < d$.

## 2.3 Adaptive Active Latent Dimensions

Thus far our analysis has been predicated on the existence of a single $r$-dimensional manifold underlying the data $x$, along with a conditioning variable $c$ that captures $t$ degrees-of-freedom within this manifold. More broadly though, it is reasonable to envision scenarios whereby the data instead lie on a union of manifolds, each with a locally-defined value of $r$ and possibly $t$ for continuous conditioning variables. In such instances, both Theorems 1 and 2 can be naturally refined to reflect this additional flexibility.

While we defer a formal treatment to future work, it is nonetheless worthwhile to consider CVAE behavior within two representative scenarios. First, consider the case where $c$ is now a *discrete* random variable taking a value in some set/alphabet $\{\alpha_k\}_{k=1}^m$, such as the label of an MNIST digit [23] (whereby $m = 10$). It then becomes plausible to assume that $r = f(\alpha_k)$ for some function $f$, meaning that the manifold dimension itself may vary conditioned on the value of $c$ (e.g., the space of digits "1" is arguably simpler than the space of digits "8"). In principle then, a suitably-designed CVAE model trained on such data should be able to adaptively learn the dimensions of these regional manifolds. Later in Section 4.4 we empirically demonstrate that when we include a specialized attention layer within the CVAE decoder, which allows it to selectively shut on and off different latent dimensions, the resulting model can indeed learn the underlying union of low-dimensional manifolds under certain circumstances. From a conceptual standpoint, the outcome is loosely analogous to a separate, class-conditional VAE being trained on data associated with each $\alpha_k$.

As a second scenario, we may also consider the case where $t$ varies for different values of a *continuous* conditioning variable $c$. Extrapolating from Theorem 2, we would expect that the number of active latent dimensions $r - t$ will now vary across regions of the data space. Hence an appropraite CVAE architecture should be able to adaptively compress the active latent dimensions so as to align with the varying information contained in $c$. Again, Section 4.4 demonstrates that this is indeed possible.

## 3 On Common CVAE Model Design Choices

In this section, we review CVAE model designs and training practices that, while adopted in various prior CVAE use cases, nonetheless may have underappreciated consequences, especially within the present context of learning the underlying data manifold dimensionality.

### 3.1 On the Equivalence of Conditional and Unconditional Priors

Per the canonical CVAE design, it is common to include a parameterized, trainable prior $p_\theta(z|c)$ within CVAE architectures [1, 7, 8, 20, 24]. However, the strict necessity of doing so is at least partially compromised by the following remark:

**Remark 1 (Converting conditional to unconditional priors)** *Consider a $\kappa$-simple CVAE model with prior $p_\theta(z|c)$, encoder $q_\phi(z|x,c)$ and decoder $p_\theta(x|z,c)$. We can always find another $\kappa$-simple CVAE model with prior $p(z) \sim \mathcal{N}(0, I)$, encoder $q_{\phi'}(z|x,c)$ and decoder $p_{\theta'}(x|z,c)$, such that $\mathcal{L}(\theta, \phi) = \mathcal{L}(\theta', \phi')$ and $p_\theta(x|c) = p_{\theta'}(x|c)$.*

Remark 1 indicates that, at least in principle, a parameterized, conditional prior is not unequivocally needed. Specifically, as detailed in the appendix, we can always explicitly convert an existing $\kappa$-

simple CVAE model with conditional prior $p_\theta(z|c)$ into another $\kappa$-simple CVAE model with fixed prior $p(z) = \mathcal{N}(z|0, I)$ without sacrificing any model capacity in the resulting generative process $p_{\theta'}(x|c) = \int p_{\theta'}(x|z, c)p(z)dz$; essentially the additional expressivity of the conditional prior is merely absorbed into a revised decoder. Even so, there may nonetheless remain differences in the optimization trajectories followed during training such that achievable local minima may at times lack this equivalence.

### 3.2 The Impact of $\gamma$ Initialization on Model Convergence

As emphasized previously, VAE/CVAE models with sufficient capacity applied to manifold data achieve the minimizing loss when $\gamma \to 0$. However, directly fixing $\gamma$ near zero can be problematic for reasons discussed in [3], and more broadly, performance can actually be compromised when $\gamma$ is set to any fixed positive constant as noted in [4, 10, 15, 18].

Even so, it remains less well-understood how the initializaton of a learnable $\gamma$ may impact the optimization trajectory during training. After all, the analysis we have provided is conditioned on finding global solutions, and yet it is conceivable that different $\gamma$ initializations could influence a model's ability to steer around bad local optima. Note that the value of $\gamma$ at the beginning of training arbitrates the initial trade-off between the reconstruction and KL terms, as well as the smoothness of the initial loss landscape, both of which are factors capable of influencing model convergence. We empirically study these factors in Section 4.5.

### 3.3 A Problematic Aspect of Encoder/Prior Model Weight Sharing

Presumably to stabilize training and/or avoid overfitting, one widely-adopted practice is to share CVAE weights between the prior and posterior/encoder modules [8, 17, 19]. For generic, fixed-sized input data this may take the form of simply constraining the encoder as $q_\phi(z|x, c) = p_\theta(z|c)$ [17]. More commonly though, for sequential data both prior and encoder are instantiated as some form of recurrent network with shared parameters, where the only difference is the length of the input sequence [8, 19], i.e., the full sequence for the encoder or the partial conditioning sequence for the prior.

More concretely with respect to the latter, assume sequential data $x = \{x_l\}_{l=1}^n$, where $l$ is a time index (for simplicity here we will assume a fixed length $n$ across all samples, although this can be easily generalized). Then associated with each time point $x_l$ within a sample $x$, we have a prior conditioning sequence $c_l = x_{<l} \triangleq \{x_j\}_{j=1}^{l-1}$. The resulting encoder and prior both define distributions over a corresponding latent $z_l$ via $q_\phi(z_l|x_{\leq l})$ and $p_\theta(z_l|x_{<l})$ respectively. Along with the decoder model $p_\theta(x_l|z_l, x_{<l})$, the revised, sequential CVAE training loss becomes

$$\mathcal{L}(\theta, \phi) = \int_\mathcal{X} \sum_{l=1}^n \{-\mathbb{E}_{q_\phi(z_l|x_{\leq l})} \left[\log p_\theta(x_l|z_l, x_{<l}) + \mathbb{KL}\left[q_\phi(z_l|x_{\leq l})||p_\theta(z|x_{<l})\right]\}\omega_{gt}(dx), \quad (3)$$

where we observe that there is now an additional summation over the temporal index $l$. This mirrors the fact that to reconstruct (or at test time generate) a sample $x$, the CVAE will sequentially produce each time point $x_l$ conditioned on previously reconstructed (or generated) values. We now analyze one of the underappreciated consequences of encoder/prior weight-sharing within the aforementioned sequential CVAE context.

**Theorem 3 (Weight sharing can compromise the performance of sequential CVAEs)** *Assume sequential data with ground-truth measure $\omega_{gt}$ defined such that the probability mass of $x_l$ conditioned on $x_{<l}$ lies on a manifold with minimum dimension $r > 1$ (this excludes strictly deterministic sequences). Moreover, given a $\kappa$-simple sequential CVAE model,[3] assume that the prior is constrained to share weights with the encoder such that $p_\theta(z_l|x_{<l}) = q_\phi(z_l|x_{<l})$. Then the corresponding loss from (3) satisfies $\mathcal{L}(\theta, \phi) = \Omega(1)$ for any $\theta$ and $\phi$.*

Given the rigidity of this lower bound, Theorem 3 indicates that the sequential CVAE model is unable to drive the the loss towards minus infinity as required to produce active latent dimensions (per Definition 5) and correctly learn low-dimensional manifolds when present. Intuitively, this relates to subtleties of the differing roles the encoder and prior play in sequence models.

---

[3]This is defined analogously to the $\kappa$-simple CVAE from Definition 4.

In particular, it follows that the encoder distribution $q_\phi(z_l|x_{\leq 1})$ at time $l$ is in every way *identical* to the prior at time $l+1$ given that $p_\theta(z_{l+1}|x_{<l+1}) = q_\phi(z_{l+1}|x_{<l+1}) = q_\phi(z_{l+1}|x_{\leq l})$ per the adopted weight sharing assumption. But these distributions are meant to serve two very different purposes: (i) The *encoder* is meant to push the variances of active dimensions to zero so as to learn the underlying manifold, while the remaining/superfluous dimensions merely output useless noise that is filtered by the decoder. (ii) In stark contrast, the role of the *prior* is not to instantiate reconstructions, but rather to inject the calibrated randomness needed to match the conditional uncertainty in the ground-truth distribution. Therefore, unless the observed data sequences are deterministic, in which case $x_l$ can be predicted exactly from $x_{<l}$, it is *impossible* to achieve both (i) and (ii) simultaneously. And indeed, we verify these insights via numerical experiments conducted in Section 4.6.

## 4   Experiments

In this section we first corroborate our previous analysis in a controllable, synthetic environment; later we extend to real-world datasets to further support our conclusions.[4] The VAE/CVAE model architectures used for our experiments are described in Section A of the appendix.

### 4.1   Datasets and Metrics

**Synthetic Data**   We begin by generating low-dimensional data $u \in \mathbb{R}^r$ from a Gaussian mixture model with 5 equiprobability components, each with mean $\mu_u$ and transformed variance $\log \sigma_u$ drawn from a standardized Gaussian distribution. Next, to produce the high-dimensional data manifold $\mathcal{X} = \mathcal{M}_r$ in $\mathbb{R}^d$, we compute $x = \text{sigmoid}(G_x u)$, where $G_x \in \mathbb{R}^{d \times r}$ is a randomly initialized projection matrix. For conditional models, we also compute $c = G_c u_{1:t}$, where $G_c \in \mathbb{R}^{t \times t}$ is another random projection. Given these assumptions, we can then construct training datasets with varying values of $r$, $d$, and $t$. For all experiments with synthetic data, the training set size is 100,000. We also perform tests using synthetic sequential data described in Section 4.6.

**Real-world Data**   We also investigate model behaviors via MNIST [23] and Fashion MNIST [22].[5] These two datasets involve image samples with clear-cut low-dimensional manifold structure, while at the same time, they are not so complex that more intricate architectures and training designs are required that might otherwise obfuscate our intended message (e.g., more complex models may fail at times to learn correct low-dimensional structure simply because of convergence issues).

**Metrics**   By convention, we report the loss from (1) for VAEs and (2) for CVAEs; these losses are often referred to as the negative evidence lower bound (-ELBO) (note though that these values are not always directly comparable across different testing scenarios). We also include auxiliary metrics to diagnose model behavior, including the number of active dimensions (AD), the reconstruction error (Recon), the KL-divergence ($\mathbb{KL}$), and the learned decoder variance $\gamma$.

### 4.2   Learning Manifold Dimensions with VAEs

We begin by exploring the VAE's ability to learn the correct manifold dimension in support of Theorem 1. Specifically, using the synthetic dataset, we vary the ambient dimension $d \in \{10, 20, 30\}$ and the ground-truth manifold dimension $r \in \{2, 4, 6, 8, 10\}$ and compare with the estimated number of active dimensions with $\kappa = \in \{5, 20\}$. Please see Section B of the appendix for details regarding how active dimensions are computed in practice (while in numerical experiments encoder variances will not converge to exactly zero, we can nonetheless observe these variances closely clustering around either 0 or 1 as expected).

Results are shown in Table 1, where we observe that when $\kappa \geq r$, the VAE can generally align the active dimensions with the value of $r$ as expected once $\gamma$ has converged to a small value. In contrast, when $\kappa$ is too small (i.e., $\kappa = 5$), it is no longer possible to learn the manifold dimension (not surprisingly, the -ELBO and reconstruction error are also much larger as well).

Complementary results on MNIST and Fashion MNIST are reported in Table 2. While we no longer have access to the ground-truth value of $r$, we can still observe that as $\kappa$ increases, the number of active dimensions saturates as expected.

---

[4]Code is available at https://github.com/zhengyjzoe/manifold-dimensions-cvae
[5]MNIST is under the CC BY-SA 3.0 license, and Fashion MNIST is MIT-Licensed.

Table 1: Aligning learned VAE active dimensions (AD) with the ground-truth manifold dimension $r$. When $\kappa \geq r$ (a surplus of latent dimensions) the VAE largely succeeds as $\gamma$, the reconstruction error (Recon), and -ELBO converge to relatively small values; however, when $\kappa < r$ this is not possible.

| $\kappa$ | $d$ | $r$ | AD | Recon | $\mathbb{KL}$ | $\gamma$ | -ELBO |
|---|---|---|---|---|---|---|---|
| | | 2 | 2 | $3\times10^{-4}$ | 18.31 | $1.625\times10^{-5}$ | -58.26 |
| | | 4 | 4 | $2.6\times10^{-3}$ | 24.22 | $5.654\times10^{-5}$ | -29.83 |
| | 10 | 6 | 6 | $9.2\times10^{-3}$ | 24.14 | $3\times10^{-4}$ | -17.39 |
| | | 8 | 7 | $1.27\times10^{-2}$ | 27.91 | $1.4\times10^{-3}$ | -10.38 |
| | | 10 | 8 | $5.99\times10^{-2}$ | 16.39 | $2.5\times10^{-3}$ | -6.40 |
| | | 2 | 2 | $1.6\times10^{-3}$ | 17.98 | $5.052\times10^{-5}$ | -114.52 |
| | | 4 | 4 | $1.75\times10^{-2}$ | 23.11 | $2\times10^{-4}$ | -60.90 |
| 20 | 20 | 6 | 6 | $3.09\times10^{-2}$ | 28.96 | $6\times10^{-4}$ | -43.75 |
| | | 8 | 8 | $3.42\times10^{-2}$ | 33.83 | $1.2\times10^{-3}$ | -36.82 |
| | | 10 | 10 | $4.74\times10^{-2}$ | 35.81 | $1.1\times10^{-3}$ | -28.34 |
| | | 2 | 2 | $2.6\times10^{-3}$ | 18.42 | $7.221\times10^{-5}$ | -176.74 |
| | | 4 | 4 | $2.73\times10^{-2}$ | 24.60 | $2\times10^{-4}$ | -100.28 |
| | 30 | 6 | 6 | $4.74\times10^{-2}$ | 31.89 | $9\times10^{-4}$ | -76.46 |
| | | 8 | 8 | $5.68\times10^{-2}$ | 37.28 | $1.6\times10^{-3}$ | -65.66 |
| | | 10 | 10 | $1.13\times10^{-1}$ | 35.13 | $2.5\times10^{-3}$ | -47.00 |
| | | 6 | 5 | $1.299\times10^{-1}$ | 22.53 | $2.1\times10^{-3}$ | -36.97 |
| 5 | 20 | 8 | 5 | $3.719\times10^{-1}$ | 16.618 | $8.8\times10^{-3}$ | -22.60 |
| | | 10 | 5 | $3.564\times10^{-1}$ | 15.966 | $1.113\times10^{-2}$ | -16.96 |

Table 2: VAE latent compression on real datasets in further support of Theorem 1.

| Dataset | $\kappa$ | AD | Recon | -ELBO |
|---|---|---|---|---|
| | 5 | 5 | 14.899 | -842.286 |
| MNIST | 16 | 12 | 9.749 | -1065.83 |
| | 32 | 13 | 7.469 | -1224.37 |
| | 5 | 5 | 13.163 | -935.127 |
| Fashion MNIST | 16 | 9 | 9.026 | -1216.68 |
| | 32 | 9 | 7.820 | -1327.26 |

### 4.3 Learning Manifold Dimensions with CVAEs

Turning to CVAEs, we provide empirical support for Theorem 2 using synthetic datasets with varying $t$, the effective dimension of the conditioning variable $c$. For this purpose, we set $c$ to be the first $t$ dimensions of $u$ by equating $G_c$ to a $t$-dim identity matrix $I_t$, and let $d = \kappa = 20$, and $r = 10$. Table 3 shows the results, whereby the CVAE correctly learns that $AD = r - t$ across all values of $t$.

Meanwhile, on MNIST and Fashion MNIST we train CVAEs with the class label of each image as the conditioning variable, and the results with $\kappa = 32$ are in Table 4. By comparing with Table 2, we observe that the CVAE model for MNIST has a lower AD and -ELBO, suggesting that when conditioned on labels, the model can more easily confine the resulting representations to a lower-dimensional manifold. In contrast, for FashionMNIST the CVAE/VAE results are more similar, indicating that the class labels provide marginal benefit, possibly because their visual complexity and manifold structure is more similar across classes. We also provide an example of the CVAE encoder variances produced on the MNIST dataset in Section B of the appendix.

### 4.4 Adaptive CVAE Active Dimensions within a Dataset

Continuing with CVAE experimentation, we now turn to verifying aspects of Section 2.3, namely, the ability to adaptively learn active dimensions that vary regionally within a single dataset composed of a union of low-dimensional ground-truth manifolds. To this end, we choose $c$ as a *discrete* indicator

Table 3: CVAE latent compression showing AD $= r - t$ exactly in support of Theorem 2. Here $c$ is the first $t$ dimensions of $u$, i.e. $G_c = I_t$, $d = \kappa = 20$, and $r = 10$.

| $t$ | -ELBO | Recon | $\mathbb{KL}$ | $\gamma$ | AD |
|---|---|---|---|---|---|
| 1 | -31.41 | $4.61 \times 10^{-2}$ | 33.26 | $2.4 \times 10^{-3}$ | 9 |
| 3 | -36.67 | $4.66 \times 10^{-2}$ | 27.78 | $2.4 \times 10^{-3}$ | 7 |
| 5 | -42.78 | $4.86 \times 10^{-2}$ | 20.81 | $2.6 \times 10^{-3}$ | 5 |
| 7 | -52.39 | $4.29 \times 10^{-2}$ | 13.72 | $2.2 \times 10^{-3}$ | 3 |
| 9 | -62.25 | $3.84 \times 10^{-2}$ | 6.07 | $2 \times 10^{-3}$ | 1 |

Table 4: CVAE latent compression in real datasets with $\kappa = 32$. AD is averaged over all the classes.

| | AD | Recon | $\mathbb{KL}$ | $\gamma$ | -ELBO |
|---|---|---|---|---|---|
| MNIST | 12 | 6.044 | 81.672 | 0.0063 | -1489.42 |
| Fashion MNIST | 9 | 8.773 | 54.552 | 0.0102 | -1239.09 |

of each source manifold, 5 in total with $r \in \{1, ..., 5\}$ latent dimensions within each respectively. Table 5 shows the results using $d = 20$ and $\kappa = 40$, noting that both the AD and -ELBO values are class-conditional within a single dataset. Also, we report performance both with and without a special attention layer in the decoder that helps to selectively determine which dimensions should be active on a sample-by-sample basis. These results indicate that, when equipped with a suitably flexible decoder network, the CVAE has the ability to adaptively learn AD values that correctly align with data lying on a union of manifolds.

To further explore adaptive active dimensions, we also consider the case where $c$ is a continuous variable with $t \in \{2, 4, 6, 8, 10\}$ varying from sample-to-sample (this is achieved by using the first $\{2, 4, 6, 8, 10\}$ dimensions of $u$ as the conditioning variable). To maintain a constant size of $c \in \mathbb{R}^{10}$ we apply zero-padding. Using $r = 12$, $d = 20$, and $\kappa = 20$, results are shown in Table 6, again with and without the attention layer that exploits $c$ to correctly learn the optimal AD across all values of $t$.

### 4.5 The Impact of $\gamma$ Initialization on Model Training

We next investigate how the initialization of $\gamma$ impacts VAE/CVAE model convergence as related to the discussion in Section 3.2. Results with synthetic data and $d = 20$, $\kappa = 20$, $r = 10$, and for the CVAE, $t = 5$ are shown in Table 7 as the initial $\gamma$ is varied. Here we observe that for the VAE model, the correct number of active dimensions is only learned when $\gamma$ is initialized sufficiently small. In contrast, for the CVAE we include results both with and without a parameterized prior. However, given an adequately-sized decoder, both models perform similarly as would be expected (note that we used the same decoder for both models, so the parameter-free prior model was technically not strictly equivalent per Remark 1). And with the exception of the $\log \gamma = 20$ case when the parameterized CVAE prior model falls into a local minima, they are both able to correctly learn the number of active CVAE dimensions as AD $= r - t = 5$, consistent with Theorem 2. In comparison with the VAE, the CVAE models are likely less sensitive to the initial $\gamma$ because the number of active latent dimensions that need to be learned is half that of the VAE (5 versus 10).

Table 5: Adaptively learning CVAE active dimension involving data lying on a union of 5 manifolds with $r \in \{1, \ldots, 5\}$, and $d = 20, \kappa = 40$. A discrete $c$ labels each manifold/class, and the AD values and -ELBO are computed on a class-conditional basis.

| $r$ | True AD | AD without attention | AD with attention | -ELBO without attention | -ELBO with attention |
|---|---|---|---|---|---|
| 1 | 1 | 4 | 1 | -102.25 | -114.22 |
| 2 | 2 | 4 | 2 | -62.42 | -99.81 |
| 3 | 3 | 4 | 3 | -60.13 | -74.28 |
| 4 | 4 | 4 | 4 | -28.06 | -50.36 |
| 5 | 5 | 4 | 5 | -7.58 | -59.25 |

Table 6: Adaptively learning active dimensions involving a continuous $c$ and associated $t \in \{2, 4, 6, 8, 10\}$ varying within a single dataset; also $r = 12, d = 20, \kappa = 90$. The AD values and -ELBO are computed on a class-conditional basis.

| $t$ | True AD | AD without attention | AD with attention | -ELBO without attention | -ELBO with attention |
|----|----|----|----|----|----|
| 2 | 10 | 10 | 10 | -9.69 | -41.49 |
| 4 | 8 | 10 | 8 | -33.71 | -20.52 |
| 6 | 6 | 10 | 6 | -27.10 | -73.26 |
| 8 | 4 | 10 | 4 | -50.70 | -80.64 |
| 10 | 2 | 10 | 2 | -61.77 | -55.14 |

Table 7: Learned active dimensions as initial $\gamma$ is varied, with $d = 20, r = 10, t = 5, \kappa = 20$.

| Init $\log \gamma$ | VAE | | CVAE $p(z)$ | | CVAE $p_\theta(z|c)$ | |
|----|----|----|----|----|----|----|
| | AD | -ELBO | AD | -ELBO | AD | -ELBO |
| -20 | 10 | -28.39 | 5 | -41.20 | 5 | -40.72 |
| -10 | 9 | -28.57 | 5 | -44.53 | 5 | -45.25 |
| 0 | 8 | -27.56 | 5 | -44.38 | 5 | -45.2 |
| 10 | 3 | -13.89 | 5 | -43.72 | 5 | -43.66 |
| 20 | 1 | -1.7 | 5 | -45.22 | 4 | -37.85 |

## 4.6 Performance Degradation Using Shared Weights between Encoder and Prior

Finally, we provide empirical support for Theorem 3 relating to the potential impact of weight-sharing within sequential models. We generate sequences via an autoregressive-moving-average (ARMA) process, and we define the conditioning variable $c_l$ at time $l$ as the concatenation of the previous 5 timepoints within each sequence. Results training a CVAE model to generate analogous sequences are shown in Table 8. We observe that when the encoder and prior share parameters (first row) the -ELBO cannot be significantly reduced, largely because the reconstruction error remains high. In contrast, without shared weights (second row) the model successfully drives the reconstruction error and -ELBO to much smaller values as might be expected based on Theorem 3. Of course in larger, more complex models there could be other mitigating factors, such as overfitting risks, that might allow shared weights to at times perform relatively better.

Table 8: Impact of weight sharing between encoder and prior on sequential data.

| Shared Weights | -ELBO | Recon | $\mathbb{KL}$ | $\gamma$ |
|----|----|----|----|----|
| True | -2.49 | 0.374 | 18.09 | 0.012 |
| False | -45.015 | $1.81 \times 10^{-5}$ | 175.99 | $7.252 \times 10^{-7}$ |

## 5 Conclusion

In this paper we provide several insights into the behavior of VAEs and CVAEs applied to data lying on low-dimensional manifolds. This includes a formal demonstration that both VAE and CVAE global minima can learn manifold dimensions underlying the data, including those manifolds that have been modulated by conditioning variables. We also explore common CVAE design choices that can have practical implications when applying to various specialized applications, such as sequential data or data composed of a union of manifolds.

**Limitations** Our primary theoretical contributions are predicated on properties of VAE/CVAE global optima assuming sufficient model capacity to represent ground-truth manifolds. However, we have spoken relatively little about strategies for steering optimization trajectories away from bad local solutions that may have poor, unpredictable properties. Similarly, beyond that attention mechanism we mentioned for learning region-specific active latent dimensions, we have not thoroughly explored what types of decoder inductive biases might best align with real-world manifold data.

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
