# Appendix

## A  Details of Model Architectures and Implementation

**Synthetic Data**  For the experiments with VAE models, both the encoder and decoder are defined as a Multi Layer Perceptron (MLP) with a single hidden layer. For all the experiments with CVAE models except Section 4.6, the encoder first processes conditioning variable $c$ via an MLP, then concatenates the output and samples $x$ as the input of another MLP. The decoder processes $c$ in the same way and then uses an MLP to decode the latent variable. In Section 4.4, the attention layer we use in the decoder is a trainable vector which is applied as the weight of the latent vector at the top layer. In Section 4.6, both the encoder and decoder are LSTMs with one hidden layer.

**Real Data**  Both the encoder and decoder use two ResNet blocks to process MNIST/ Fashion MNIST images. Each encoder block is a residual network which contains two $3 \times 3$ Conv-BN-ReLU modules in its main branch and one $1 \times 1$ Conv-BN module in its shortcut. The decoder block contains a single-layer ConvNet residual block followed by a ConvTranspose layer.

**Resources**  We conduct our experiments on an Amazon Web Services g4dn.12xlarge EC2 instance, which provides 4 T4 GPUs. We estimate that the time to run through all experiments in this paper once would cost 20 GPU-hours. The research activity for this paper cost around 100 GPU-hours in total.

## B  Encoder Variance Illustration

To show the active dimensions visually, here we report the encoder variances both on synthetic data and the MNIST dataset. Note that when the value of the encoder variance is less than $0.05$, we categorize the corresponding dimension as active for VAE models; for CVAEs we analogously require that the encoder/prior variance ratio is less than $0.05$. Note however in Table 1 and 2 below, the number of active dimensions is quite obvious given the clear clustering of variance values.

Table 1: VAE encoder variance matrix on synthetic data associated with Table 1 of the main text, where $\kappa = 20, d = 30, r = 6$ and we find the number of active dimensions is 6. The estimated active dimensions are in blue.

| | | | | |
|---|---|---|---|---|
| 0.0080 | 0.0018 | 1.0000 | 1.0000 | 1.0000 |
| 0.0027 | 0.0031 | 1.0000 | 1.0000 | 1.0000 |
| 1.0000 | 0.0087 | 1.0000 | 0.0141 | 1.0000 |
| 1.0000 | 1.0000 | 1.0000 | 1.0000 | 1.0000 |

Table 2: CVAE Encoder variance matrix of the CVAE model on MNIST dataset from Table 4 of the main text, where $\kappa = 32$ and the number of active dimensions is 12. The estimated active dimensions are labeled in blue.

| | | | |
|---|---|---|---|
| 3.6159e-03 | 9.6320e-01 | 7.6566e-04 | 3.5173e-04 |
| 9.8518e-01 | 9.6739e-01 | 9.6077e-01 | 8.1020e-04 |
| 9.8065e-01 | 9.7336e-01 | 3.7781e-03 | 7.1394e-04 |
| 9.6985e-01 | 6.1294e-03 | 9.7449e-01 | 9.8012e-01 |
| 7.8233e-04 | 9.7318e-01 | 9.8596e-01 | 2.4359e-04 |
| 9.7785e-01 | 9.7737e-01 | 9.7315e-01 | 9.8431e-01 |
| 9.2616e-01 | 9.8335e-01 | 9.6775e-01 | 1.2756e-03 |
| 1.0324e-03 | 9.6723e-01 | 9.6046e-01 | 2.1289e-03 |

## C  Proof of Theorem 1

**Summary of the Proof**  We define three categories based on the number of active dimensions and the rate of their encoder variance. Note that any possible VAE optimum has to fall into one of the following three categories: the number of dimensions whose encoder variance $\sigma_z^2(x, \phi_\gamma^*) = O(\gamma)$ is either greater than $r$, equal to $r$ or less than $r$. The proof's logic flow is:

1. When the number of active dimensions whose encoder variance $\sigma_z^2(x) = O(\gamma)$ is less than $r$, the reconstruction error will increase at a rate of $O(\frac{1}{\gamma})$, thus the cost cannot reach the optimum. This is proven in Section C.1;

2. When the stated dimension number equals $r$, the optimal cost is exactly $(d-r)\log\gamma + O(1)$. The corresponding proof is in Section C.2;

3. When the stated dimension number is greater than $r$, denoted as $m > r$, the cost is $(d-m)\log\gamma + O(1) > (d-r)\log\gamma + O(1)$ as shown in Section C.3.

The $O(\gamma)$ rate of the reconstruction error also follows naturally from these results.

## C.1 The number of active dimensions whose encoder variance $\sigma_z^2(x) = O(\gamma)$ is less than $r$

The main idea is to link the gap between a large $\sigma_z$ and large reconstruction error. For a given $z_0$, $\mu_x(z_0)$ will equal some $x_0$ such that $||x_0 - \mu_x(z_0)||^2 = 0$. But for other choices from $\mathcal{X}$ where $x \neq x_0$, we have $||x - \mu_x(z_0)||^2 > 0$ leading to the positive expectation term $\int_z q(z|x)||x - \mu_x(z)||^2 dz$. To minimize such positive error, we need to lower the density $q(z|x)$ where $x \neq x_0$, which is a function of $\sigma_z$.

Suppose that the number of active dimensions whose encoder variance $\sigma_z^2(x) = O(\gamma)$ is less than $r$. In this section, we will show that under this assumption the model can't reach its global optimum, i.e. $\mathcal{L} \nrightarrow -\infty$. Remind that the cost of VAE is

$$\mathcal{L}(\theta, \phi) = \int_{\mathcal{X}} \{-\mathbb{E}_{q_\phi(z|x)}[\log p_\theta(x|z)] + \mathbb{KL}[q_\phi(z|x)||p(z)]\}\omega_{gt}(dx)$$

We have

$$2\mathcal{L}(\theta, \phi) = \int_{\mathcal{X}} \{-2\mathbb{E}_{q_\phi(z|x)}[\log p_\theta(x|z)] + 2\mathbb{KL}[q_\phi(z|x)||p(z)]\}\omega_{gt}(dx)$$

$$= d\log(2\pi\gamma) + \int_{\mathcal{X}} \{\gamma^{-1}\mathbb{E}_{q_\phi(z|x)}[||x - \mu_x(z)||^2] + 2\mathbb{KL}(q_\phi(z|x)||p(z))\}\omega_{gt}(dx)$$

$$= d\log(2\pi\gamma) + \gamma^{-1}\int_{\mathcal{X}}\int_z q_\phi(z|x)[||x - \mu_x(z)||^2]dz \,\omega_{gt}(dx)$$

$$+ \int_{\mathcal{X}} 2\mathbb{KL}(q_\phi(z|x)||p(z))\omega_{gt}(dx)$$

(1)

Following the two facts:

1. Lebesgue measure on the real numbers is $\sigma$-finite.

2. $z \in \mathbb{R}^\kappa$ and $x \in \mathcal{X}$, where $\mathcal{X}$ is a $r$-dimensional manifold embedded in $\mathbb{R}^d$.

and referring to Fubini's theorem, we can switch the integration order of $\omega_{gt}(dx)$ and $dz$. Assume the components of $z \in \mathcal{Z}^\kappa \subseteq \mathbb{R}^\kappa$ is permutable. For a $r$-dimensional manifold, we can always use the first $r$ dimensions of $z$ to get $\varphi(x)$, i.e. once given r-dimensional information, there always exists a decoder, s.t. $\mu_x(z_{1:r}) = \mu_x(z)$. Denote by $\mu_z(x)_{1:r}$ and $\sigma_z^2(x)_{1:r}$ the mean and covariance matrix of the first $r$ dimension of $z$. After switching the integration order, we have

$$\int_{\mathcal{X}} \frac{1}{\gamma} \int_z q_\phi(z|x)[||x - \mu_x(z)||^2]dz \,\omega_{gt}(dx) + \int_{\mathcal{X}} [d\log(2\pi\gamma) + 2\mathbb{KL}(q_\phi(z|x)||p(z))]\,\omega_{gt}(dx)$$

$$= \frac{1}{\gamma} \int_z \int_{\mathcal{X}} q_\phi(z|x)[||x - \mu_x(z_{1:r})||^2]\omega_{gt}(dx)dz + \int_{\mathcal{X}} [d\log\gamma + 2\mathbb{KL}(q_\phi(z|x)||p(z)) + O(1)]\,\omega_{gt}(dx)$$

$$= \frac{1}{\gamma} \int_{z \in \mathcal{Z}^r} \int_{\mathcal{X}} \frac{1}{\sqrt{(2\pi)^r|\sigma_z^2(x)_{1:r}|}} e^{-\frac{1}{2}(z - \mu_z(x)_{1:r})^T \sigma_z^{-2}(x)_{1:r}(z - \mu_z(x)_{1:r})}[||x - \mu_x(z)||^2]\omega_{gt}(dx)dz +$$

$$\int_{\mathcal{X}} [d\log\gamma + 2\mathbb{KL}(q_\phi(z|x)||p(z)) + O(1)]\,\omega_{gt}(dx)$$

(2)

**C.1.1   Analyze the density with respect to $\sigma_z(x)_{1:r}$ and $z_{1:r} - \mu_z(x)_{1:r}$**

Next, for the integral over $\mathcal{X}$ in the first term in (2), we examine a certain $z_{1:r} \in \mathcal{Z}^r$ and view it as a constant. Since $\mu_x$ is a deterministic function, $\mu_x(z_{1:r})$ is also constant. The log-density on $z_{1:r}$ is

$$\frac{r}{2}\log(\frac{1}{2\pi}) + \frac{1}{2}\log\frac{1}{|\sigma_z^2|} - \frac{1}{2}(z_{1:r} - \mu_z(x)_{1:r})^T \sigma_z^{-2}(z_{1:r} - \mu_z(x)_{1:r})$$

Take the derivative of $\sigma_z^2$, we have

$$-\frac{\sigma_z^{-2}}{2} + \frac{1}{2}\sigma_z^{-2}(z_{1:r} - \mu_z(x)_{1:r})(z_{1:r} - \mu_z(x)_{1:r})^T \sigma_z^{-2}$$

When $\sigma_z^2$ is smaller than $(z_{1:r} - \mu_z(x)_{1:r})(z_{1:r} - \mu_z(x)_{1:r})^T$, the second term's rate is larger. Thus the density is monotonically increasing when $\sigma_z^2 \prec (z_{1:r} - \mu_z(x)_{1:r})(z_{1:r} - \mu_z(x)_{1:r})^T$ and monotonically decreasing when $\sigma_z^2 \succ (z_{1:r} - \mu_z(x)_{1:r})(z_{1:r} - \mu_z(x)_{1:r})^T$. Note that $\mu_x$ is $L$-Lipschitz continuous, so we have $L|z_{1:r} - \mu_z(x)_{1:r}| \geq |\mu_x(z_{1:r}) - \mu_x(\mu_z(x)_{1:r})| = |\mu_x(z_{1:r}) - x|$. The equality comes from the fact that we can choose optimal $\mu_z$ and $\mu_x$, s.t. $\mu_x(\mu_z(x)_{1:r}) = x$.

Now we can divide $x \in \mathcal{X}$ into four cases and we assume all the four disjoint cases exist when analyzing, otherwise the integration over corresponding domain is 0 which would not affect our result. The four cases are as follows:

1. $\mathcal{X}_1(z_{1:r}) = \{x : \sigma_z^2(x)_{1:r} \prec (z_{1:r} - \mu_z(x)_{1:r})(z_{1:r} - \mu_z(x)_{1:r})^T\} \cap \{x : ||z_{1:r} - \mu_z(x)_{1:r}|| = +\infty\}$

2. $\mathcal{X}_2(z_{1:r}) = \{x : \sigma_z^2(x)_{1:r} \prec (z_{1:r} - \mu_z(x)_{1:r})(z_{1:r} - \mu_z(x)_{1:r})^T\} \cap \{x : ||z_{1:r} - \mu_z(x)_{1:r}|| < +\infty\}$

3. $\mathcal{X}_3(z_{1:r}) = \{x : (z_{1:r} - \mu_z(x)_{1:r})(z_{1:r} - \mu_z(x)_{1:r})^T \preceq \sigma_z^2(x)_{1:r} < \infty\}$

4. $\mathcal{X}_4(z_{1:r}) = \{x : (z_{1:r} - \mu_z(x)_{1:r})(z_{1:r} - \mu_z(x)_{1:r})^T \preceq \sigma_z^2(x)_{1:r} = \infty\}$

We have $\mathcal{X}_1(z_{1:r}) \cup \mathcal{X}_2(z_{1:r}) = \{x : \sigma_z^2(x)_{1:r} \prec (z_{1:r} - \mu_z(x)_{1:r})(z_{1:r} - \mu_z(x)_{1:r})^T\}$ and $\mathcal{X}_3(z_{1:r}) \cup \mathcal{X}_4(z_{1:r}) = \{x : \sigma_z^2(x)_{1:r} \succeq (z_{1:r} - \mu_z(x)_{1:r})(z_{1:r} - \mu_z(x)_{1:r})^T\}$. Thus $\mathcal{X}_1(z_{1:r}) \cup \mathcal{X}_2(z_{1:r}) \cup \mathcal{X}_3(z_{1:r}) \cup \mathcal{X}_4(z_{1:r})$ cover the whole space of $x$ related to $z_{1:r}$, i.e. $\mathcal{X}(z_{1:r})$.

*(i) $\mathcal{X}_1(z_{1:r}) = \{x : \sigma_z^2(x)_{1:r} \prec (z_{1:r} - \mu_z(x)_{1:r})(z_{1:r} - \mu_z(x)_{1:r})^T\} \cap \{x : ||z_{1:r} - \mu_z(x)_{1:r}|| = +\infty\}$*

Denote $\sigma_l^2$ as the lower bound of $\sigma_z^2$'s eigenvalues, which cannot approach 0 by our assumption. $\sigma_z < +\infty$. The integral over $\mathcal{X}_1(z_{1:r})$

$$\int_{\mathcal{X}_1(z_{1:r})} \frac{1}{\sqrt{|\sigma_z^2|}} e^{-\frac{1}{2}(z_{1:r} - \mu_z(x)_{1:r})^T \sigma_z^{-2}(z_{1:r} - \mu_z(x)_{1:r})}[||x - \mu_x(z_{1:r})||^2]\omega_{gt}(dx)_r$$

$$\leq \int_{\mathcal{X}_1(z_{1:r})} \frac{1}{\sigma_l^r} e^{-\frac{1}{2}(z_{1:r} - \mu_z(x)_{1:r})^T \sigma_z^{-2}(z_{1:r} - \mu_z(x)_{1:r})}[L^2||z_{1:r} - \mu_z(x)_{1:r}||^2]\omega_{gt}(dx)_r$$

will approach 0 as $||z_{1:r} - \mu_z(x)_{1:r}|| = +\infty$. Thus $\int_{z_{1:r}} 0 dz = 0$

*(ii) $\mathcal{X}_2(z_{1:r}) = \{x : \sigma_z^2(x)_{1:r} \prec (z_{1:r} - \mu_z(x)_{1:r})(z_{1:r} - \mu_z(x)_{1:r})^T\} \cap \{x : ||z_{1:r} - \mu_z(x)_{1:r}|| < +\infty\}$*

Denote $N = \max_x\{||z_{1:r} - \mu_z(x)_{1:r}||^2\}$ and $\mathcal{X}_2^\alpha(z_{1:r}) = \{x : ||x - \mu_x(z_{1:r})||^2 > \alpha\} \cap \mathcal{X}_2(z_{1:r})$, where $\alpha \nrightarrow 0$. If for any $\alpha$, $\mathcal{X}_2^\alpha(z_{1:r}) = \emptyset$, we have for all $x \in \mathcal{X}_2(z_{1:r})$, $x = \mu_x(z_{1:r})$. However, if $\mu_x(x)_r \in \mathcal{X}_2(z_{1:r})$, i.e. $\mu_x(z_{1:r})$ satisfies $\sigma_z^2(\mu_x(z_{1:r})) \prec (z_{1:r} - \mu_z(\mu_x(z_{1:r})))(z_{1:r} - \mu_z(\mu_x(z_{1:r})))^T$. We can find a pair of $\mu_x$ and $\mu_z$, e.g. identity mapping, s.t. $\mu_z(\mu_x(z_{1:r})) = z_{1:r}$ and $\sigma_z^2(\mu_x(z_{1:r})) < 0$, which is impossible. Thus $\mu_x(z_{1:r}) \notin \mathcal{X}_2(z_{1:r})$ and $\mathcal{X}_2(z_{1:r}) = \emptyset$. Thus, once $\mathcal{X}_2(z_{1:r}) \neq \emptyset$, there exists an $\alpha$, s.t. $\mathcal{X}_2^\alpha(z_{1:r}) \neq \emptyset$. The integral over $\mathcal{X}_2(z_{1:r})$

$$\int_{\mathcal{X}_2(z_{1:r})} \frac{1}{\sqrt{|\sigma_z^2|}} e^{-\frac{1}{2}(z_{1:r}-\mu_z(x)_{1:r})^T \sigma_z^{-2}(z_{1:r}-\mu_z(x)_{1:r})} [||x-\mu_x(z_{1:r})||^2] \omega_{gt}(dx)_r$$

$$\geq \int_{\mathcal{X}_2(z_{1:r})} \frac{1}{\sigma_l^r} e^{-\frac{1}{2}\sigma_l^{-2r}N} [||x-\mu_x(z_{1:r})||^2] \omega_{gt}(dx)_r$$

$$= \frac{1}{\sigma_l^r} e^{-\frac{1}{2}\sigma_l^{-2r}N} [\int_{\mathcal{X}_2^\alpha(z_{1:r})} [||x-\mu_x(z_{1:r})||^2] \omega_{gt}(dx)_r + \int_{(\mathcal{X}_2^\alpha(z_{1:r}))^c} [||x-\mu_x(z_{1:r})||^2] \omega_{gt}(dx)_r]$$

$$\geq \frac{\alpha}{\sigma_l^r} e^{-\frac{1}{2}\sigma_l^{-2r}N}$$

The last inequality comes from the fact that $\varpi(\mathcal{X}_2^\alpha(z_{1:r})) \geq 1$ where $\varpi$ is a counting measure and the non-negativity of the second term.

*(iii)* $\mathcal{X}_3(z_{1:r}) = \{x : (z_{1:r}-\mu_z(x)_{1:r})(z_{1:r}-\mu_z(x)_{1:r})^T \preceq \sigma_z^2(x)_{1:r} < \infty\}$

In this case the density is monotonically decreasing with $\sigma_z$. Since $\sigma_z \neq +\infty$, denote $\sigma_u^2$ as the upper bound of the eigenvalues of $\sigma_z^2$. It can also bound $||z_{1:r}-\mu_z(x)_{1:r}||^2$. Use the same strategy in *(ii)*, define $\mathcal{X}_3^{\alpha'}(z_{1:r}) = \{x : ||x-\mu_x(z_{1:r})||^2 > \alpha'\} \cap \mathcal{X}_3(z_{1:r})$. If $\mathcal{X}_3(z_{1:r}) \neq \emptyset$, we have

$$\int_{\mathcal{X}_3(z_{1:r})} \frac{1}{\sqrt{|\sigma_z^2|}} e^{-\frac{1}{2}(z_{1:r}-\mu_z(x)_{1:r})^T \sigma_z^{-2}(z_{1:r}-\mu_z(x)_{1:r})} [||x-\mu_x(z_{1:r})||^2] \omega_{gt}(dx)_r$$

$$\geq \int_{\mathcal{X}_3(z_{1:r})} \frac{1}{\sigma_u^r} e^{-\frac{r}{2}\sigma_u^{-2r+2}} [||x-\mu_x(z_{1:r})||^2] \omega_{gt}(dx)_r$$

$$= \frac{1}{\sigma_u^r} e^{-\frac{r}{2}\sigma_u^{-2r+2}} [\int_{\mathcal{X}_3^{\alpha'}(z_{1:r})} [||x-\mu_x(z_{1:r})||^2] \omega_{gt}(dx)_r + \int_{(\mathcal{X}_3^{\alpha'}(z_{1:r}))^c} [||x-\mu_x(z_{1:r})||^2] \omega_{gt}(dx)_r]$$

$$\geq \frac{\alpha'}{\sigma_u^r} e^{-\frac{r}{2}\sigma_u^{-2r+2}}$$

*(iv)* $\mathcal{X}_4(z_{1:r}) = \{x : (z_{1:r}-\mu_z(x)_{1:r})(z_{1:r}-\mu_z(x)_{1:r})^T \preceq \sigma_z^2(x)_{1:r} = \infty\}$

In this case the density is monotonically decreasing with $\sigma_z$, and the dominant factor is $\frac{1}{\sqrt{|\sigma_z^2(x)_{1:r}|}}$. Since $\sigma_z$ is arbitrarily large, it is obvious that $\sqrt{|\sigma_z^2|} > tr(\sigma_z^2) \geq ||z_{1:r}-\mu_z(x)_{1:r}||^2 \geq \frac{1}{L^2}||x-\mu_x(z_{1:r})||^2$. Note that $|\cdot| = det(\cdot)$.

The integral over $\mathcal{X}_4(z_{1:r})$

$$\int_{\mathcal{X}_4(z_{1:r})} \frac{1}{\sqrt{|\sigma_z^2|}} e^{-\frac{1}{2}(z_{1:r}-\mu_z(x)_{1:r})^T \sigma_z^{-2}(z_{1:r}-\mu_z(x)_{1:r})} [||x-\mu_x(z_{1:r})||^2] \omega_{gt}(dx)_r$$

$$\leq \int_{\mathcal{X}_4(z_{1:r})} \frac{L^2||z_{1:r}-\mu_z(x)_{1:r}||^2}{\sqrt{|\sigma_z^2(x)_{1:r}|}} \omega_{gt}(dx)_r$$

will approach 0 as $\sigma_z^2 \to \infty$.

### C.1.2 Analyze the existence of the above cases and get a lower bound

We have $\mathcal{X} = \mathcal{X}_1 \cup \mathcal{X}_2 \cup \mathcal{X}_3 \cup \mathcal{X}_4$, where $\mathcal{X}_i = \cup_{z_{1:r}} \mathcal{X}_i(z_{1:r})$, $i = 1, 2, 3, 4$. In $\mathcal{X}_1 \cup \mathcal{X}_4$, the integral is 0. To get a lower bound of (2), we need to prove $\mathcal{X}_2 \cup \mathcal{X}_3 \neq \emptyset$, i.e. there must exists $z_{1:r}$ such that $x \in \{\sigma_z^2(x)_{1:r} < \infty\} \cap \{||z_{1:r}-\mu_z(x)_{1:r}|| < \infty\}$ exists.

For $\sigma_z(x)_r$, if $\sigma_z(x)_r = \infty$, then in the KL term the trace $tr(\sigma_z^2(x)_{1:r}) = +\infty$ which cannot be offset by $-\log|\sigma_z^2(x)_{1:r}|$. Thus to minimize loss, $\sigma_z < \infty$.

For $||z_{1:r}-\mu_z(x)_{1:r}|| < \infty$, with $L$-Lipschitz continuity, for any $z_{1:r}^*$, we can find a $x^* \in \mathcal{X}$, s.t. $||z_{1:r}^* - \mu_z(x^*)_{1:r}|| = 0$. Denote $U_\delta(x)_r$ as a neighborhood of $x$ with the radius of $\delta$. For any $x \in U_\delta(x^*)$, we have

$$||\mu_z(x)_{1:r} - z^*_{1:r}|| = ||\mu_z(x)_{1:r} - \mu_z(x^*)_{1:r}|| \le L||x - x^*|| \le L\delta$$

So $U_\delta(x^*) \subset \mathcal{X}_2 \cup \mathcal{X}_3$. To get a positive lower bound, we need to prove there exists $x'$ and $\delta$, s.t. the image of $\mu_z(x')$ for $x \in U_\delta(x')$ is with positive measure. If for any $x \in U_\delta(x^1)$, $\mu_z(x)_{1:r} = z^1_{1:r}$, and for any $x \in U_\delta(x^2)$, $\mu_z(x)_{1:r} = z^2_{1:r}$, which satisfy $\delta < ||x^2 - x^1|| \le \frac{3}{2}\delta$, and $z^1_{1:r} \ne z^2_{1:r}$. Note that can always choose a larger $\delta$ to get such pair of $\{x^1, x^2\}$. Then there exists $x^3 \in U_\delta(x^1) \cap U_\delta(x^2)$, $\mu_z(x^3)$ should equals $z^1_{1:r}$ and $z^2_{1:r}$ simultaneously which is impossible. Thus, there must exists $x^*$, s.t. $\mu_z(U_\delta(x^*))$ has a positive measure.

With the existence of $x^*$, such that $U_\delta(x^*) \subset \mathcal{X}_2 \cup \mathcal{X}_3$ and positive measured $\mu_z(U_\delta(x^*))$, we have

$$
\begin{aligned}
&\frac{1}{\gamma} \int_{z_{1:r}} \int_{\mathcal{X}} \frac{1}{\sqrt{(2\pi)^r |\sigma_z^2|}} e^{-\frac{1}{2}(z_{1:r} - \mu_z(x)_{1:r})^T \sigma_z^{-2}(z_{1:r} - \mu_z(x)_{1:r})} [||x - \mu_x(z_{1:r})||^2] \omega_{gt}(dx)_r dz_{1:r} + \\
&\int_{\mathcal{X}} [d\log\gamma + 2\mathbb{KL}(q_\phi(z|x)||p(z)) + O(1)] \omega_{gt}(dx) \\
=&\frac{1}{\gamma} \int_{z_{1:r}} \int_{\mathcal{X}_2 \cup \mathcal{X}_3} \frac{1}{\sqrt{(2\pi)^r |\sigma_z^2|}} e^{-\frac{1}{2}(z_{1:r} - \mu_z(x)_{1:r})^T \sigma_z^{-2}(z_{1:r} - \mu_z(x)_{1:r})} [||x - \mu_x(z_{1:r})||^2] \omega_{gt}(dx)_r dz_{1:r} + \\
&\int_{\mathcal{X}} [d\log\gamma + 2\mathbb{KL}(q_\phi(z|x)||p(z)) + O(1)] \omega_{gt}(dx) \\
\ge&\frac{1}{\gamma} \int_{z_{1:r}} \min\{\frac{\alpha}{\sigma_l^r} e^{-\frac{1}{2}\sigma_l^{-2r} N}, \frac{\alpha'}{\sigma_u^r} e^{-\frac{r}{2}\sigma_u^{-2r+2}}\} dz_{1:r} + \\
&\int_{\mathcal{X}} [d\log\gamma + 2\mathbb{KL}(q_\phi(z|x)||p(z)) + O(1)] \omega_{gt}(dx) \\
=&\frac{C}{\gamma} + \int_{\mathcal{X}} [d\log\gamma - \log|\sigma_z^2(x)| + O(1)] \omega_{gt}(dx)
\end{aligned}
$$
$$(3)$$

Here denote $C = \int_{z_{1:r}} \min\{\frac{\alpha}{\sigma_l^r} e^{-\frac{1}{2}\sigma_l^{-2r} N}, \frac{\alpha'}{\sigma_u^r} e^{-\frac{r}{2}\sigma_u^{-2r+2}}\} dz_{1:r}$ for simplicity.

### C.1.3 Analyze the rate of the lower bound

The first term $\frac{C}{\gamma}$ grows at a rate of $O(\frac{1}{\gamma})$. Because $\sigma_z^2$ is at a lower rate than $\gamma$, we have

$$O(d\log\gamma - \log|\sigma_z^2(x)|) < O(-(d - \kappa)\log\frac{1}{\gamma})$$

and the right part decreases at a rate of $\log\frac{1}{\gamma}$. When $\gamma \to 0$, $O(\frac{1}{\gamma}) > O(\log\frac{1}{\gamma})$, which means the increase from reconstruction term cannot be offset by the decrease from the KL term. Moreover, from the fact that $O(\frac{1}{\gamma}) > O(\log\frac{1}{\gamma})$, when $\gamma$ is small enough, the loss is monotonically decreasing with $\gamma$.

Therefore, when $\gamma \to 0$, the lower bound cannot approach $-\infty$, which means at this case, the model can never achieve optimum. Thus, there must exist some active dimensions whose variance $\sigma_z^2(x)$ satisfies $\sigma_z^2(x) = O(\gamma)$ as $\gamma \to 0$ to reach the global optimum. We can learn from the expression of $C$ that as long as the number of such active dimensions whose encoder variance $\sigma_z^2(x) = O(\gamma)$ exceeds $r$, as $\gamma$ approaches zero, the reconstruction term is at most at the rate of $O(1)$. Next, we will show that when there exist at least $r$ such active dimensions, the VAE model's optimum is achievable.

### C.2 The number of active dimensions whose encoder variance $\sigma_z^2(x) = O(\gamma)$ equals $r$

In this section, we get an upper bound and a lower bound and show that both case the cost is $(d - r)\log\gamma + O(1)$.

### C.2.1 An Upper Bound of ELBO

**Get an upper bound by Lipschitz** We can write $z = \mu_z(x) + \varepsilon * \sigma_z(x)$, where $\varepsilon \sim N(0, I)$. Since decoder mean function $\mu_x(z; \theta)$ is $L$-Lipschitz continuous, we have:

$$\mathbb{E}_{z \sim q_{\phi_\gamma}(z|x)}[||x - \mu_x(z)||^2]$$
$$= \mathbb{E}_{\varepsilon \sim N(0,I)}[||\mu_x(\mu_z(x)_{1:r}) - \mu_x(z_{1:r})||^2]$$
$$\leq \mathbb{E}_{\varepsilon \sim N(0,I)}[||L(\mu_z(x)_{1:r} - \mu_z(x)_{1:r} - \sigma_z(x)_{1:r}\varepsilon)||^2] \quad (4)$$
$$= \mathbb{E}_{\varepsilon \sim N(0,I)}[||L\sigma_z(x)_{1:r}\varepsilon||^2]$$

where the first equality comes from the fact that we can choose optimal encoder-decoder pairs such that $\mu_x(\mu_z(x)_{1:r}) = x$. Take it into $\mathcal{L}$,

$$2\mathcal{L}(\sigma_z(x)_{1:r}, \gamma)$$
$$= \int_{\mathcal{X}} \left[ \mathbb{E}_{z \sim q_\phi(z|x)}[\frac{1}{\gamma}||x - \mu_x(z)||_2^2] + d\log 2\pi\gamma - \log|\sigma_z^2(x)_{1:r}| - \log|\sigma_z^2(x)_{r+1:\kappa}| + O(1) \right] \omega_{gt}(dx)$$
$$\leq \frac{L^2}{\gamma} \int_{\mathcal{X}} \left[ \mathbb{E}_{\varepsilon \sim N(0,I)}[||\sigma_z(x)_{1:r}\varepsilon||^2] + d\log 2\pi\gamma - \log|\sigma_z^2(x)_{1:r}| - \log|\sigma_z^2(x)_{r+1:\kappa}| + O(1) \right] \omega_{gt}(dx)$$
$$(5)$$

We get an upper bound of $\mathcal{L}$, denoted as $\tilde{\mathcal{L}}$.

**Analysis of the Upper Bound $\tilde{\mathcal{L}}$** Now we only pay attention to the upper bound $\tilde{\mathcal{L}}$ and try to prove that it is at a rate of $O((d - r)\log\gamma)$.

We can get implicit optimal values of $\tilde{\mathcal{L}}$: $\gamma^*$ and $\sigma^*(x)_{1:r}^2$ by taking the derivative of $\tilde{\mathcal{L}}$ separately.

We have optimal $\gamma^*$

$$\gamma^* = \arg\min_\gamma \tilde{\mathcal{L}}(\theta, \phi) = \frac{L^2}{d}\mathbb{E}_{\varepsilon \sim N(0,I)}[||\sigma_z(x)_{1:r}\varepsilon||^2] \quad (6)$$

and

$$\frac{\partial 2\tilde{\mathcal{L}}(\sigma_z(x)_{1:r}, \gamma)}{\partial \sigma_z(x)_{1:r}} = \frac{2L^2\sigma_z(x)_{1:r}}{\gamma}\mathbb{E}_{\varepsilon \sim N(0,I)}[\varepsilon\varepsilon^T] - 2\sigma_z(x)_{1:r}^{-1}$$
$$= \frac{2L^2\sigma_z(x)_{1:r}}{\gamma} - 2\sigma_z(x)_{1:r}^{-1} = 0$$

we have the optimal variance of $\tilde{\mathcal{L}}$:

$$\sigma_z^*(x)_{1:r}^2 = \gamma\frac{I}{L^2} \quad (7)$$

It shows that $\frac{1}{\sqrt{\gamma}}\sigma_z^*(x)_{1:r} = O(1)$ when it reaches the optimal value.

Take the optimal values into $\tilde{\mathcal{L}}$, then we get $\tilde{\mathcal{L}}$ as a function of $\gamma^*$ and $\sigma_z^*(x)_{1:r}$:

$$2\tilde{\mathcal{L}}(\gamma^*, \sigma_z^*(x)_{1:r})$$
$$= \int_{\mathcal{X}} \left[ \frac{L^2}{\gamma^*}\mathbb{E}_{\varepsilon \sim N(0,I)}[||\sigma_z(x)_{1:r}\varepsilon||^2] + d\log 2\pi\gamma^* - \log|\sigma_z^*(x)_{1:r}^2| - \log|\sigma_z(x)_{r+1:\kappa}^2| + O(1) \right] \omega_{gt}(dx)$$
$$= \int_{\mathcal{X}} \left[ d + d\log(2\pi\gamma^*) - \log|\gamma^*\frac{I}{L^2}| - \log|\sigma_z(x)_{r+1:\kappa}^2| + O(1) \right] \omega_{gt}(dx)$$
$$= d\log(2\pi\gamma^*) - r\log\gamma^* - \log|\sigma_z(x)_{r+1:\kappa}^2| + O(1)$$
$$(8)$$

Define $\{\lambda_i\}_{i=1}^\kappa$ as the eigenvalues of $\sigma_z(x)$. Denote $\tilde{r}$ as the number of $\{\lambda_i\}_{i=r+1}^\kappa$ that will go to 0 as $\gamma^* \to 0$. We have

$$2\tilde{\mathcal{L}}(\gamma^*) = d\log\gamma^* - r\log\gamma^* - 2\sum_{i=r+1}^{r+\tilde{r}}\log\lambda_i - 2\sum_{i=\tilde{r}+r+1}^{\kappa}\log\lambda_i + O(1) \tag{9}$$

To minimize (9), we want $\tilde{r}$ to be as small as possible and at the best equals 0 which is achievable. Since the rest $\kappa - r - \tilde{r} = \kappa - r$ dimensions are irrelevant to $\gamma$, at least will not approach 0 when $\gamma \to 0$, we can view them as constants. We have the loss equals

$$(d-r)\log\gamma + O(1) \tag{10}$$

### C.2.2 A Lower Bound of ELBO

From C.1 we have analyzed the loss performance when there are less than $r$ active dimensions whose variance goes to zero at a rate no lower than $\gamma$. In this part, we focus on the case that $r$ latent dimensions are such active dimensions whose encoder variance goes to zero at a rate of $O(\gamma)$. Without loss of generality, we assume the first $r$ latent dimensions satisfy $\sigma_z^2(x)_{1:r} = O(\gamma)$ as $\gamma \to 0$. We have

$$
\begin{aligned}
2\mathcal{L}(\theta,\phi) &= \int_{\mathcal{X}}\{-2\mathbb{E}_{q_\phi(z|x)}[\log p_\theta(x|z)] + 2\mathbb{KL}[q_\phi(z|x)||p(z)]\}\omega_{gt}(dx)\\
&= \int_{\mathcal{X}}\left\{\frac{1}{\gamma}\mathbb{E}_{q_\phi(z|x)}[||x-\mu_x(z)||^2] + d\log(2\pi\gamma) + 2\mathbb{KL}(q_\phi(z|x)||p(z))\right\}\omega_{gt}(dx)\\
&\geq \int_{\mathcal{X}}\left\{d\log(2\pi\gamma) - \log|\sigma_z^2(x)| + O(1)\right\}\omega_{gt}(dx)\\
&= \int_{\mathcal{X}}\left\{d\log\gamma - \log|\sigma_z^2(x)_{1:r}| - \log|\sigma_z^2(x)_{r+1:\kappa}| + O(1)\right\}\omega_{gt}(dx)\\
&\geq \int_{\mathcal{X}}\left\{(d-r)\log\gamma - \log|\sigma_z^2(x)_{r+1:\kappa}| + O(1)\right\}\omega_{gt}(dx)\\
&= (d-r)\log\gamma + O(1)
\end{aligned}
\tag{11}
$$

The inequalities come from the fact that the norm term is non-negative and the active dimensions' rate is no less than $\gamma$. For the last equality, we can use the strategy in (9). To minimize the lower bound, there should not be any active dimensions in these $r+1 : \kappa$ dimensions.

We get an upper bound and a lower bound at the same rate, i.e. $\log\gamma$ with $r$ active latent dimensions. Therefore, the original loss is also with $r$ active dimensions. We have the optimal cost for each $x$ equals

$$(d-r)\log\gamma + O(1) \tag{12}$$

So far, we have get the conclusion in Theorem 1 about the form of ELBO when $\gamma \to 0$, as well as the number and rate of active dimensions. Next, we show that the number of active dimensions can't be greater than $r$.

### C.3 When the number of active dimensions is greater than $r$

Denote now there are $m$ active dimensions and $m > r$. From (8), in this case $\tilde{r} = m - r$, and the loss is

$$\frac{1}{\gamma}\mathbb{E}_{q_{\phi_\gamma}(z|x)}[||x-\mu_x(z)||^2] + d\log(2\pi\gamma) - 2\sum_{i=1}^{r}\log\lambda_i - 2\sum_{i=r+1}^{r+\tilde{r}}\log\lambda_i + O(1) \tag{13}$$

since here $\lim_{\gamma\to 0}\lambda_i = 0$, $-2\sum_{i=r+1}^{r+\tilde{r}}\log\lambda_i$ is monotonically increases as $\tilde{r}$ increases at the rate of $\Omega(\log\frac{1}{\gamma})$. For the reconstruction term, it is unaffected since we only use the first $r$ latent dimensions

for reconstruction. Therefore, the loss will increase at the rate of $\Omega(\log \frac{1}{\gamma})$, which is larger than the loss when $m = r$.

In conclusion, only when the number of active dimensions equals $r$, and these active dimensions' encoder variance $\sigma_z^2(x) = O(\gamma)$ as $\gamma \to 0$, the optimal cost is $(d - r) \log \gamma + O(1)$.

# D   Proof of Theorem 2

**Summary of the proof**   We focus on analyzing the loss conditioned on a specific $c$, which is defined as $\mathcal{L}_c(\theta, \phi)$. We then first construct the proof when $p_\theta(z|c) = p(z)$, i.e. a parameter free prior, and then extend to the case when the prior involves the conditioning variable. The logic flow is as follows:

1. The prior is independent of $c$

    (a) Following the same proof idea as in Theorem 1, when the number of active dimensions whose encoder variance $\sigma_z^2(x, c; \phi) = O(\gamma)$ is less than $r - k$, where $k$ is the number of effective dimensions of $c$ used in the decoder, the reconstruction error will grow at a rate of $O(\frac{1}{\gamma})$. This is proven in Section D.1.1;

    (b) In Section D.1.2 we show that both the upper bound and the lower bound are $(d - r + t) \log \gamma$ and the exact number of active dimensions is $r - t$.

2. The prior is a function of $c$

    (a) Since involving $c$ in the prior will not affect the reconstruction term, we extend the conclusion in Section D.1.1 to the general case;

    (b) Show that both the upper bound and the lower bound are $(d - r + t) \log \gamma$ and the exact number of active dimensions is $r - t$. The proof is in Section D.2.

Under CVAE setting, we first make some denotations for proof. Since the encoder, prior, and decoder share the same condition $c$, the model has flexibility to use part of each $c$ from the three networks. Denote $t$ as the number of effective dimensions of $c$, and $k$ as the number of effective dimensions of $c$ used in the decoder $p_\theta(x|z, c)$, i.e. there exists a pair of encoder and decoder, s.t. $\mu_x(c) = \varphi^{-1}(u_{1:k})$, where $0 \le k \le t$ and is a learnable parameter. The encoder and prior use the rest $t - k$ effective dimensions, i.e. $\mu_x(\mu_z(c)) = \varphi^{-1}(u_{k+1:t})$, and this part of information will be included in the latent variable $z$.

## D.1   When the prior is independent of $c$, i.e. $p_\theta(z|c) = p(z)$

In this case, we can write the cost as:

$$
\begin{aligned}
2\mathcal{L}_c(\theta, \phi) =& 2 \int_{\mathcal{X}_c} \{-\mathbb{E}_{q_\phi(z|x,c)}[\log p_\theta(x|z,c)] + \mathbb{KL}[q_\phi(z|x,c)||p(z)]\} \omega_{gt}^c(dx) \\
=& \int_{\mathcal{X}_c} \frac{1}{\gamma} \mathbb{E}_{q_\phi(z|x,c)}[||x - \mu_x(z,c)||^2] + d \log(2\pi\gamma) + 2\mathbb{KL}(q_\phi(z|x,c)||p(z)) \omega_{gt}^c(dx) \\
=& \frac{1}{\gamma} \int_{\mathcal{X}_c} \mathbb{E}_{q_{\phi_\gamma}(z|x,c)}[||\varphi^{-1}(u_{1:k}) - \mu_x(c)||^2 + ||\varphi^{-1}(u_{k+1:r}) - \mu_x(z_{k+1:r})||^2 + \\
& d \log(2\pi\gamma) + 2\mathbb{KL}(q_\phi(z|x,c)||p(z)) \omega_{gt}^c(dx) \\
=& \frac{1}{\gamma} \int_{\mathcal{X}_c} \mathbb{E}_{q_{\phi_\gamma}(z|x,c)}[||\varphi^{-1}(u_{k+1:r}) - \mu_x(z_{k+1:r})||^2 + d \log(2\pi\gamma) + 2\mathbb{KL}(q_\phi(z|x,c)||p(z)) \omega_{gt}^c(dx)
\end{aligned}
$$

## D.1.1   The number of active dimensions whose encoder variance $\sigma_z^2(x, c; \phi) = O(\gamma)$ is less than $r - k$

Following the same proof idea in Theorem 1, assume there is no active dimension in $\sigma_z(x; \phi)$. For the reconstruction term, it is equivalent to reconstruct a $(r - k)$-dimensional manifold. Thus we can find an lower bound of the cost

$$2\mathcal{L}_c(\theta,\phi) \geq \frac{C'}{\gamma} + \int_{\mathcal{X}_c} \left[d\log\gamma + 2\mathbb{KL}(q_\phi(z|x,c)||p(z)) + O(1)\right]\omega_{gt}^c(dx)$$

$$= \frac{C'}{\gamma} + \int_{\mathcal{X}_c} \left[d\log\gamma - \log|\sigma_z^2(x,c;\phi)_{1:k}| - \log|\sigma_z^2(x,c;\phi)_{k+1:\kappa-t+k}| + O(1)\right]\omega_{gt}^c(dx) \tag{14}$$

where $C' = \int_{z_{1:r-k}} \min\{\frac{\alpha}{\sigma_l^{r-k}}e^{-\frac{1}{2}\sigma_l^{-2(r-k)}}N, \frac{\alpha'}{\sigma_u^{r-k}}e^{-\frac{r-k}{2}\sigma_u^{-2(r-k)+2}}\}dz_{1:r-k}$.

The first term $\frac{C'}{\gamma}$ grows at a rate of $O(\frac{1}{\gamma})$. Because $\sigma_z^2$ is at a lower rate than $\gamma$, we have

$$O(d\log\gamma - \log|\sigma_z^2(x,c;\phi)|) < O(-(d-\kappa)\log\frac{1}{\gamma})$$

and the right part decreases at a rate of $\log\frac{1}{\gamma}$. When $\gamma \to 0$, $O(\frac{1}{\gamma}) > O(\log\frac{1}{\gamma})$, which means the increase from reconstruction term cannot be offset by the decrease from the KL term. Moreover, from the fact that $O(\frac{1}{\gamma}) > O(\log\frac{1}{\gamma})$, when $\gamma$ is small enough, the loss is monotonically decreasing with $\gamma$.

Therefore, when $\gamma \to 0$, the lower bound cannot approach $-\infty$, which means at this case, the model can never achieve optimum. Thus, there must exist some active dimensions whose variance satisfies $\sigma_z^2(x,c;\phi)_i = O(\gamma), i = 1,\ldots,\kappa$ as $\gamma \to 0$ to reach the global optimum, and it is showed in $C'$ that as long as the number of such active dimensions exceeds $r-k$, as $\gamma$ approaches zero, the reconstruction term is at most at the rate of $O(1)$. Next, we will show that when there exist at least $r-k$ such active dimensions, the CVAE model's optimum is achievable.

### D.1.2 Bounds of CVAE cost

**The upper bound** We have $z_{1:t-k} = \mu_z(x,c;\phi)_{1:t-k} + \sigma_z(x,c;\phi)_{1:t-k}\varepsilon_1$ and $z_{t-k+1:r} = \mu_z(x,c;\phi)_{t-k+1:r-k} + \sigma_z(x;\phi)_{t-k+1:r-k}\varepsilon_2$, where $\varepsilon_1 \sim N(0,I^{t-k}), \varepsilon_2 \sim N(0,I^{r-t})$.

The loss is:

$$\frac{1}{\gamma}\mathbb{E}_{q_{\phi_\gamma}(z|x,c)}[||x - \mu_x(z,c)||^2] + d\log(2\pi\gamma) + 2\mathbb{KL}(q_\phi(z|x,c)||p(z))$$

$$= \frac{1}{\gamma}\mathbb{E}_{q_{\phi_\gamma}(z|x,c)}[||\varphi^{-1}(u_{1:k}) - \mu_x(c)||^2 + ||\varphi^{-1}(u_{k+1:t}) - \mu_x(z_{1:t-k})||^2 +$$

$$||\varphi^{-1}(u_{t+1:r}) - \mu_x(z_{t-k+1:r-k})||^2] + d\log(2\pi\gamma) + 2\mathbb{KL}(q_\phi(z|x,c)||p(z))$$

$$\leq \frac{1}{\gamma}\mathbb{E}_{\varepsilon_1 \sim N(0,I^{t-k})}[||L\sigma_z(x,c;\phi)_{1:t-k}\varepsilon_1||^2] + \frac{1}{\gamma}\mathbb{E}_{\varepsilon_2 \sim N(0,I^{r-t})}[||L\sigma_z(x,c;\phi)_{t-k+1:r-k}\varepsilon_2||^2] +$$

$$d\log\gamma - \log|\sigma_z^2(x,c;\phi)_{1:t-k}| - \log|\sigma_z^2(x,c;\phi)_{t-k+1:r-k}| + O(1) \tag{15}$$

Denote $\sigma_z^2(x,c;\phi)_{1:t-k}$ as $\sigma_z^2(c,\phi)$ for simplicity, and denote the upper bound of loss as $\mathcal{L}_c^u$. Take the derivative of $\sigma_z(c;\phi)$ and $\sigma_z(x,c;\phi)_{t-k+1:r-k}$ separately. Because the diagonal elements in $\sigma_z(x,c;\phi)$ are independent, we can make both achieve optimum. We have:

$$\mathcal{L}_c^u(\gamma,k) = -(t-k)\log\gamma - (r-t)\log\gamma + d\log\gamma + O(1) \tag{16}$$

To minimize $\mathcal{L}_c^u(\gamma,k)$, the optimal $k$ is $t$, thus the upper bound is:

$$\mathcal{L}_c^u(\gamma) = (d-r+t)\log\gamma + O(1) \tag{17}$$

**The lower bound** We have show that there must be at least $r-k$ active dimension at a rate of $O(\gamma)$, otherwise the loss will increase at a rate of $O(\gamma)$. We can get a lower bound

$$\frac{1}{\gamma}\mathbb{E}_{q_{\phi_\gamma}(z|x,c)}[||x - \mu_x(z,c)||^2] + d\log(2\pi\gamma) + 2\mathbb{KL}(q_\phi(z|x,c)||p(z))$$

$$\geq d\log\gamma - \log|\sigma_z^2(x,c;\phi)_{1:r-k}| - \log|\sigma_z^2(x,c;\phi)_{r-k+1:\kappa}| + O(1) \tag{18}$$

$$\geq d\log\gamma - (r-k)\log\gamma - \log|\sigma_z^2(x,c;\phi)_{r-k+1:\kappa}| + O(1)$$

$$\geq (d - r + k)\log\gamma + O(1)$$

Denote the lower bound as $\mathcal{L}_c^l(\gamma, k)$, to minimize it, we have $k = t$, thus the lower bound is

$$\mathcal{L}_c^l(\gamma) = (d - r + t)\log\gamma + O(1)$$

Both $\mathcal{L}_c^u$ and $\mathcal{L}_c^l$ are at a rate of $O(\log\gamma)$, we come to the conclusion that the ELBO is $(d - r + t)\log\gamma + O(1)$ and the number of active dimensions is $r - t$ when $p_\theta(z|c) = p(z)$.

### D.2 The general case

Define a trainable parametric prior of $z$, i.e. $z \sim N(\mu_z(c;\theta), \sigma_z^2(c;\theta))$. Since involving $c$ in the prior doesn't affect the reconstruction term, we have the conclusion in Section D.1.1 that there are at least $r - k$ active latent dimensions at a rate of $O(\gamma)$. Without loss of generality, we assume the first $r - k$ dimension of $\sigma_z^2(x,c;\phi)$, i.e. $\sigma_z^2(x,c;\phi)_{1:r-k} = O(\gamma)$.

**The upper bound** The loss is:

$$\frac{1}{\gamma}\mathbb{E}_{q_{\phi_\gamma}(z|x,c)}[||x - \mu_x(z,c)||^2] + d\log(2\pi\gamma) + 2\mathbb{KL}(q_\phi(z|x,c)||p(z|c))$$

$$\leq \frac{1}{\gamma}\mathbb{E}_{\varepsilon_1 \sim N(0,I^{t-k})}[||L\sigma_z(c;\phi)\varepsilon_1||^2] + \frac{1}{\gamma}\mathbb{E}_{\varepsilon_2 \sim N(0,I^{r-k})}[||L\sigma_z(x,c;\phi)_{t-k+1:r-k}\varepsilon_2||^2] + d\log(2\pi\gamma) -$$

$$\log|\sigma_z^2(c,\phi)| - \log|\sigma_z^2(x,c;\phi)_{t-k+1:r-k}| - \log|\sigma_z^2(x,c;\phi)_{r-k+1:\kappa}| + \log|\sigma_z^2(c;\theta)_{1:t-k}| + \log|\sigma_z^2(c;\theta)_{t-k+1:\kappa}|$$

$$+ (\mu_{z_\phi(1:t-k)} - \mu_{z_\theta(1:t-k)})^T\sigma_z^2(c;\theta)_{1:t-k}^{-1}(\mu_{z_\phi(1:t-k)} - \mu_{z_\theta(1:t-k)})$$

$$+ (\mu_{z_\phi(t-k+1:\kappa)} - \mu_{z_\theta(t-k+1:\kappa)})^T\sigma_z^2(c;\theta)_{t-k+1:\kappa}^{-1}(\mu_{z_\phi(t-k+1:\kappa)} - \mu_{z_\theta(t-k+1:\kappa)})$$

$$- \kappa + tr(\sigma_z^2(c,\phi)/\sigma_z^2(c;\theta)_{1:t-k}) + tr(\sigma_z^2(x;\phi)/\sigma_z^2(c;\theta)_{t-k+1:\kappa}) \tag{19}$$

Since we can only control $k$ dimensions of the prior when training, take the derivative of $\mu_z(c;\theta)_{1:k}$ and $\sigma_z(c;\theta)_{1:k}$, we have

$$\mu_z(c;\theta)_{1:t-k}^* = \mu_z(c;\phi)$$

$$\sigma_z^2(c;\theta)_{1:t-k}^* = (\mu_z(c;\phi) - \mu_z(c;\theta)_{1:t-k}^*)(\mu_z(c;\phi) - \mu_z(c;\theta)_{1:t-k}^*)^T + \sigma_z^2(c,\phi) = \sigma_z^2(c,\phi) \tag{20}$$

Let them achieve the optimal values. The loss becomes

$$\frac{1}{\gamma}\mathbb{E}_{\varepsilon_1 \sim N(0,I^{t-k})}[||L\sigma_z(c;\phi)\varepsilon_1||^2] + \frac{1}{\gamma}\mathbb{E}_{\varepsilon_2 \sim N(0,I^{r-k})}[||L\sigma_z(x,c;\phi)_{t-k+1:r-k}\varepsilon_2||^2] + d\log(2\pi\gamma)$$

$$- \log|\sigma_z^2(x,c;\phi)_{t-k+1:r-k}| - \log|\sigma_z^2(x,c;\phi)_{r-k+1:\kappa}| + \log|\sigma_z^2(c;\theta)_{r-k+1:\kappa}| + tr(\sigma_z^2(x;\phi)/\sigma_z^2(c;\theta)_{k+1:\kappa})$$

$$+ (\mu_{z_\phi(t-k+1:\kappa)} - \mu_{z_\theta(t-k+1:\kappa)})^T\sigma_z^2(c;\theta)_{t-k+1:\kappa}^{-1}(\mu_{z_\phi(t-k+1:\kappa)} - \mu_{z_\theta(t-k+1:\kappa)}) + t - k - \kappa \tag{21}$$

From (21) we observe that if we have a flexible enough prior, there are $t - k$ latent dimensions that won't provide any loss both in reconstruction and kl term. To minimize (21), $\sigma_z^2(c,\phi)^* = 0$ and $\sigma_z^2(x;\phi)_{t-k+1:r-k}^* = \gamma\frac{I}{L^2}$. Let them be the optimums, and view the terms that are irrelevant with $\gamma$ when it approaches 0 as constants, we have

$$\mathcal{L}_c^{u'} = (d - r + t)\log\gamma + O(1) \tag{22}$$

**The lower bound** to get the lower bound, we have

$$\frac{1}{\gamma}\mathbb{E}_{q_{\phi_\gamma}(z|x,c)}[||x - \mu_x(z,c)||^2] + d\log(2\pi\gamma) + 2\mathbb{KL}(q_\phi(z|x,c)||p(z|c))$$

$$\geq d\log\gamma - \log|\sigma_z^2(x,c;\phi)_{t-k+1:r-k}| - \log|\sigma_z^2(x,c;\phi)_{r-k+1:\kappa}| + \log|\sigma_z^2(c;\theta)_{r-k+1:\kappa}| +$$
$$tr(\sigma_z^2(x;\phi)/\sigma_z^2(c;\theta)_{k+1:\kappa}) + (\mu_{z_\phi(t-k+1:\kappa)} - \mu_{z_\theta(t-k+1:\kappa)})^T\sigma_z^2(c;\theta)_{t-k+1:\kappa}^{-1}(\mu_{z_\phi(t-k+1:\kappa)} - \mu_{z_\theta(t-k+1:\kappa)}) + O(1)$$

$$= d\log\gamma - \log|\sigma_z^2(x,c;\phi)_{t-k+1:r-k}| + O(1)$$
$$\geq d\log\gamma - (r-t)\log\gamma + O(1)$$
$$= (d - r + t)\log\gamma + O(1)$$

$$\tag{23}$$

The last inequality comes from the conclusion that there are at least $r - k$ active dimensions at a rate of $O(\gamma)$ and the loss is monotonously increase with $\gamma$. In this case $k$ can be any integer in $[0, t]$, thus we cannot determine how many dimensions are used by the encoder and decoder separately. But no matter what value $k$ is, the cost of CVAE is

$$(d - r + t)\log\gamma + O(1)$$

In conclusion, after integrating over $\mathcal{C}$, we have

$$\mathcal{L}(\theta^*, \phi^*) = \int_\mathcal{C} \mathcal{L}_c(\theta^*, \phi^*)\nu_{gt}(dc) = (d - r + t)\log\gamma + O(1)$$

## E    Proof of Theorem 3

**Summary of the proof**    We first define a space of sequences, and then separate the sequences into two categories according to the performance of the KL term. In Section E.1, we analyze the case when the Kl term equals $O(\log\frac{1}{\gamma})$, and in Section E.2, the rate of KL term is higher than $O(\log\frac{1}{\gamma})$. In both categories, we prove that the whole cost cannot go to $-\infty$.

Let $\theta^*, \phi^* = \arg\min_{\theta,\phi} \mathcal{L}(\theta, \phi)$. Define $S \subset \mathcal{X}$ as the set of the sequences, and the sequence is defined as $\{x_l\}_{l=1}^\infty \in S$.

Consider when $l$ equals to a constant $l_0$, we have the prior as $q_{\phi^*}(z|x_{<l_0})$, and encoder as $q_{\phi^*}(z|x_{\leq l_0})$. Next, consider $l = l_0 + 1$, we have the prior as $q_{\phi^*}(z|x_{\leq l_0})$, which is exactly the same as the encoder at $l = l_0$, and the encoder as $q_{\phi^*}(z|x_{\leq l_0+1})$. The cost function at these two points are

$$\mathcal{L}_c^{(l_0)}(\theta^*, \phi^*) = -\mathbb{E}_{q_{\phi^*}(z|x_{\leq l_0})}[\log p_{\theta^*}(x_{l_0}|z, x_{<l_0})] + \mathbb{KL}[q_{\phi^*}(z|x_{\leq l_0})||q_{\phi^*}(z|x_{<l_0})]$$

and

$$\mathcal{L}_c^{(l_0+1)}(\theta^*, \phi^*) = -\mathbb{E}_{q_{\phi^*}(z|x_{\leq l_0+1})}[\log p_{\theta^*}(x_{l_0+1}|z, x_{\leq l_0})] + \mathbb{KL}[q_{\phi^*}(z|x_{\leq l_0+1})||q_{\phi^*}(z|x_{\leq l_0})]$$

respectively.

Next, we separate the sequences with varied values into **two** cases.

### E.1    KL term is at a rate of $O(\log\frac{1}{\gamma})$ when $\gamma \to 0$.

Denote $\log q_\phi(z|x_{\leq l}) - \log q_\phi(z|x_{<l}) = f_l(\gamma) = O(\log\frac{1}{\gamma})$. In this setting, we analyze the reconstruction term

$$-\mathbb{E}_{q_{\phi^*}(z|x_{\leq l_0})}[\log p_{\theta^*}(x_{l_0}|z, x_{<l_0})]$$
$$= \int_\mathcal{Z} q_{\phi^*}(z|x_{\leq l_0})\log p_{\theta^*}(x_{l_0}|z, x_{<l_0})dz$$
$$= \frac{1}{\gamma}\int_\mathcal{Z} q_{\phi^*}(z|x_{\leq l_0})[||x_{l_0} - \mu_x^*(z)||^2]dz + d\log(2\pi\gamma)$$

$$\tag{24}$$

Similarly we have

$$
\begin{aligned}
&- \mathbb{E}_{q_{\phi^*}(z|x_{\leq l_0+1})}[\log p_{\theta^*}(x_{l_0+1}|z, x_{\leq l_0})] \\
=& \frac{1}{\gamma} \int_{\mathcal{Z}} q_{\phi^*}(z|x_{\leq l_0+1})[||x_{l_0+1} - \mu_x^*(z)||^2]dz + d\log(2\pi\gamma)
\end{aligned}
\tag{25}
$$

With the condition that $\log q_\phi(z|x_{\leq l}) - \log q_\phi(z|x_{<l}) = f_l(\gamma) = O(\log\frac{1}{\gamma})$, we have

$$
\begin{aligned}
&\mathbb{KL}[q_\phi(z|x_{\leq l})||q_\phi(z|x_{<l})] \\
=& \mathbb{E}_{q_\phi(z|x_{\leq l})}[\log q_\phi(z|x_{\leq l}) - \log q_\phi(z|x_{<l})] \\
\leq& \mathbb{E}_{q_\phi(z|x_{\leq l})}[f_l(\gamma)] = f_l(\gamma)
\end{aligned}
\tag{26}
$$

That shows KL term is either small than a constant or goes to infinity at a slower rate than rate of $\log\frac{1}{\gamma}$ when $\gamma \to 0$. We can also get $q_\phi(z|x_{<l}) \geq \frac{1}{e^{f_l(\gamma)}}q_\phi(z|x_{<l})$, from which we have

$$
q_\phi(z|x_{<l}) - q_\phi(z|x_{\leq l}) \geq q_\phi(z|x_{\leq l})(\frac{1}{e^{f_l(\gamma)}} - 1)
\tag{27}
$$

Together we have

$$
\begin{aligned}
&- \mathbb{E}_{q_{\phi^*}(z|x_{\leq l_0})}[\log p_{\theta^*}(x_{l_0}|z, x_{<l_0})] - \mathbb{E}_{q_{\phi^*}(z|x_{\leq l_0+1})}[\log p_{\theta^*}(x_{l_0+1}|z, x_{\leq l_0})] \\
=& \frac{1}{\gamma}\left[\int_{\mathcal{Z}} q_{\phi^*}(z|x_{\leq l_0})||x_{l_0} - \mu_x^*(z)||^2 dz + \int_{\mathcal{Z}} q_{\phi^*}(z|x_{\leq l_0+1})||x_{l_0+1} - \mu_x^*(z)||^2 dz\right] + 2d\log(2\pi\gamma) \\
=& \frac{1}{\gamma}[\int_{\mathcal{Z}} q_{\phi^*}(z|x_{\leq l_0+1})\left[||x_{l_0} - \mu_x^*(z)||^2 + ||x_{l_0+1} - \mu_x^*(z)||^2\right] dz + \\
& \int_{\mathcal{Z}} [q_{\phi^*}(z|x_{\leq l_0}) - q_{\phi^*}(z|x_{\leq l_0+1})] ||x_{l_0} - \mu_x^*(z)||^2 dz] + 2d\log(2\pi\gamma) \\
& \int_{\mathcal{Z}} [q_{\phi^*}(z|x_{\leq l_0}) - q_{\phi^*}(z|x_{\leq l_0+1})] ||x_{l_0} - \mu_x^*(z)||^2 dz] + 2d\log(2\pi\gamma) \\
\geq& \frac{1}{\gamma}[\int_{\mathcal{Z}} q_{\phi^*}(z|x_{\leq l_0+1})\left[||x_{l_0} - \mu_x^*(z)||^2 + ||x_{l_0+1} - \mu_x^*(z)||^2\right] dz + \\
& (\frac{1}{e^{f_{l_0}(\gamma)}} - 1)\int_{\mathcal{Z}} q_\phi(z|x_{\leq l_0+1})||x_{l_0} - \mu_x^*(z)||^2]dz + 2d\log(2\pi\gamma) \\
=& \frac{1}{\gamma}\int_{\mathcal{Z}} q_{\phi^*}(z|x_{\leq l_0+1})\left[\frac{1}{e^{f_{l_0}(\gamma)}}||x_{l_0} - \mu_x^*(z)||^2 + ||x_{l_0+1} - \mu_x^*(z)||^2\right] dz + 2d\log(2\pi\gamma)
\end{aligned}
\tag{28}
$$

For any $l_0 = 1, 2, \ldots$ and all $z \in \mathcal{Z}$, we have the following cases:

1. **For any** $z \in \mathcal{Z}_1$, $\mu_x^*(z) = x_{l_0}$ **and** $\mu_x^*(z) \neq x_{l_0+1}$. We have $\frac{||x_{l_0} - \mu_x^*(z)||^2}{\gamma} \to \infty$ at a rate of $O(\frac{1}{\gamma})$.

2. **For any** $z \in \mathcal{Z}_2$, $\mu_x^*(z) = x_{l_0+1}$ **and** $\mu_x^*(z) \neq x_{l_0}$. We have $\frac{||x_{l_0+1} - \mu_x^*(z)||^2}{\gamma e^{f_{l_0}(\gamma)}} \to \infty$ at a rate of $O(\frac{1}{\gamma e^{f_{l_0}(\gamma)}})$.

3. **For any** $z \in \mathcal{Z}_3$, $\mu_x^*(z) \neq x_{l_0}$ **and** $\mu_x^*(z) \neq x_{l_0+1}$. Both cases above cause the norm term equal $\Omega(1)$.

With the setting, the lower bound of reconstruction term is

$$
\begin{aligned}
&\frac{1}{\gamma}\int_{\mathcal{Z}} q_{\phi^*}(z|x_{\leq l_0+1})\left[\frac{1}{e^{f_{l_0}(\gamma)}}||x_{l_0} - \mu_x^*(z)||^2 + ||x_{l_0+1} - \mu_x^*(z)||^2\right]dz + 2d\log(2\pi\gamma)\\
=&\frac{1}{\gamma}\int_{\mathcal{Z}_1} q_{\phi^*}(z|x_{\leq l_0+1})||x_{l_0+1} - \mu_x^*(z)||^2 dz + \frac{1}{\gamma e^{f_{l_0}(\gamma)}}\int_{\mathcal{Z}_2} q_{\phi^*}(z|x_{\leq l_0+1})||x_{l_0} - \mu_x^*(z)||^2 dz +\\
&\frac{1}{\gamma}\int_{\mathcal{Z}_3} q_{\phi^*}(z|x_{\leq l_0+1})\left[\frac{1}{e^{f_{l_0}(\gamma)}}||x_{l_0} - \mu_x^*(z)||^2 + ||x_{l_0+1} - \mu_x^*(z)||^2\right]dz + 2d\log(2\pi\gamma)\\
\geq&\frac{1}{\gamma}\int_{\mathcal{Z}_1\cup\mathcal{Z}_3} q_{\phi^*}(z|x_{\leq l_0+1})||x_{l_0+1} - \mu_x^*(z)||^2 dz + 2d\log(2\pi\gamma)
\end{aligned}
$$
(29)

Since the probability mass of $x_l$ conditioned on $x_{<l}$ lies on a manifold with at least 1 dimension, i.e. we exclude deterministic sequences, there must exist a sequence $\{x_l\}^{i_0}$, in which $\sum_{l=1}^{\infty}\int_{\mathcal{Z}} q_{\phi^*}(z|x_{\leq l+1})||x_{l+1} - \mu_x^*(z)||^2 dz \geq C$, where $C$ is a constant, otherwise all the sequences $\{x_l\}^i \in S, i = 1, 2, \ldots$ share the same values which violates our assumption.

Then for (29), there must exist a constant $C'$, such that

$$
\int_{\mathcal{Z}_1\cup\mathcal{Z}_3} q_{\phi^*}(z|x_{\leq l_0+1})||x_{l_0+1} - \mu_x^*(z)||^2 dz \geq C'
$$

Thus, the lower bound of the cost is

$$
\frac{C'}{\gamma} - 2d\log\frac{1}{2\pi\gamma}
$$

When $\gamma$ goes to zero, $O(\frac{1}{\gamma}) > O(\log\frac{1}{\gamma})$. We get the conclusion that $\mathcal{L}_c(\theta, \phi) = \int_{\mathcal{X}} \Omega(1)\omega_{gt}(dx)$ for any $\theta$ and $\phi$.

### E.2  KL term goes to infinity at a rate higher than $O(\log\frac{1}{\gamma})$.

In this case, we have

$$
\begin{aligned}
&2\mathbb{KL}[q_{\phi^*}(z|x_{\leq l_0+1})||q_{\phi^*}(z|x_{\leq l_0})]\\
=&\log\frac{|\sigma_z^2(x_{\leq l_0})|}{|\sigma_z^2(x_{\leq l_0+1})|} - \kappa + (\mu_z(x_{\leq l_0+1}) - \mu_z(x_{\leq l_0}))^T\sigma_z^{-2}(x_{\leq l_0})(\mu_z(x_{\leq l_0+1}) - \mu_z(x_{\leq l_0}))\\
&+ tr(\sigma_z^{-2}(x_{\leq l_0})\sigma_z^2(x_{\leq l_0+1}))
\end{aligned}
$$
(30)

Thus it can only happen when there are some dimensions where $\sigma_z^2(x_{\leq l_0})$ is active while $\sigma_z^2(x_{\leq l_0+1})$ is not, which indicate that $tr(\sigma_z^{-2}(x_{\leq l_0})\sigma_z^2(x_{\leq l_0+1})) \to \infty$ at a rate of $\Omega(\frac{1}{\gamma})$. We have

$$
\begin{aligned}
&-2\mathbb{E}_{q_{\phi^*}(z|x_{\leq l_0+1})}[\log p_{\theta^*}(x_{l_0+1}|z, x_{\leq l_0})] + 2\mathbb{KL}[q_{\phi^*}(z|x_{\leq l_0+1})||q_{\phi^*}(z|x_{\leq l_0})]\\
\geq&d\log(2\pi\gamma) + tr(\sigma_z^{-2}(x_{\leq l_0})\sigma_z^2(x_{\leq l_0+1})) + \log\frac{|\sigma_z^2(x_{\leq l_0})|}{|\sigma_z^2(x_{\leq l_0+1})|}\\
&-\kappa + (\mu_z(x_{\leq l_0+1}) - \mu_z(x_{\leq l_0}))^T\sigma_z^{-2}(x_{\leq l_0})(\mu_z(x_{\leq l_0+1}) - \mu_z(x_{\leq l_0}))
\end{aligned}
$$
(31)

Because $d\log(2\pi\gamma) + \log\frac{|\sigma_z^2(x_{\leq l_0})|}{|\sigma_z^2(x_{\leq l_0+1})|} \to -\infty$ at a rate of $O(\log\gamma)$ while $tr(\sigma_z^{-2}(x_{\leq l_0})\sigma_z^2(x_{\leq l_0+1})) \to \infty$ at a rate of $\Omega(\frac{1}{\gamma})$, the whole loss will go to infinity.

In summary, in both cases, when summing over $l$, $\mathcal{L}(\theta, \phi)$ will go to infinity.

## F   Justification of Remark 1

Consider a $\kappa$-simple CVAE model with encoder $q_\phi(z|x,c)$, prior $p_\theta(z|c)$ and decoder $p_\theta(x|z,c)$. Further, let $\mu_q(x,c), \sigma_q(x,c)$ be the distributional parameters for $z \sim q_\phi(z|x,c)$, $\mu_p(c), \sigma_p(c)$ be the distributional parameters for $z \sim p_\theta(z|c)$. Name this model as $M$, and we have its cost with regard to $(\theta, \phi)$ being

$$
\begin{aligned}
2\mathcal{L}_c(M;\theta,\phi) = & \int_{\mathcal{X}} \{-2\mathbb{E}_{q_\phi(z|x,c)}[\log p_\theta(x|z,c)] + 2\mathbb{KL}[q_\phi(z|x,c)||p_\theta(z|c)]\}\omega^c_{gt}(dx) \\
= & \frac{1}{\gamma} \int_{\mathcal{X}} \int_{\mathcal{Z}} \mathcal{N}(z;\mu_q(x,c),\sigma_q(x,c))||x - \mu_x(z)||^2 dz \omega^c_{gt}(dx) \\
& + \log(2\pi\gamma) + \int_{\mathcal{X}} 2\mathbb{KL}[q_\phi(z|x,c)||p_\theta(z|c)]\omega^c_{gt}(dx) \\
= & \frac{1}{\gamma} \int_{\mathcal{X}} \int_{\mathcal{Z}} \frac{1}{\sqrt{(2\pi\gamma)^d}}\exp\{-\frac{||z - \mu_q||^2}{2\sigma_q^2}\}||x - \mu_x(z)||^2 dz \omega_{gt}(dx) \\
& + \log(2\pi\gamma) + \int_{\mathcal{X}} [\log\sigma_p - \log\sigma_q - \kappa + ||\mu_q - \mu_p||^2/\sigma_p + \mathrm{tr}(\sigma_q/\sigma_p)]\omega^c_{gt}(dx)
\end{aligned}
\tag{32}
$$

Next, we construct another $\kappa$-simple CVAE, $M'$, with a standard Gaussian prior, only using computation modules in $M$. Specifically, the new prior, decoder, and encoder are defined as:

- Prior: $p'(z') = \mathcal{N}(0, \mathbf{I})$

- Decoder: $p'(x|z', c) = p_\theta(x|z' * \sigma_p(c) + \mu_p(c), c)$

- Encoder: $q'(z|x, c) = \mathcal{N}(\mu'_q, \sigma'_q)$, where

    - $\mu'_q = (\mu_q(x,c) - \mu_p(c))/\sigma_p(c)$

    - $\sigma'_q = \sigma_q(x,c)/\sigma_p(c)$

With $M'$ defined, we are going to show that it has the exact same cost value as the above one during training, i.e. $\mathcal{L}(M';\theta,\phi) = \mathcal{L}(M;\theta,\phi)$, and the generated data distribution during generation, i.e. $p'(x|z',c)\mathcal{N}(z';0,\mathbf{I}) \equiv p_\theta(x|z,c)p_\theta(z|c)$.

During training, we have $z' \sim \mathcal{N}(\mu'_q, \sigma'_q)$, thus $z' * \sigma_p(c) + \mu_p(c) \sim \mathcal{N}((\mu_q(x,c) - \mu_p(c))/\sigma_p(c) * \sigma_p(c) + \mu_p(c), \sigma_q(x,c)/\sigma_p(c) * \sigma_p(c)) = \mathcal{N}(\mu_q(x,c), \sigma_q(x,c))$. Thus, we have

$$
\begin{aligned}
& \mathbb{E}_{q(z'|x,c)}[\log p_\theta(x|z' * \sigma_p(c) + \mu_p(c), c)] \\
= & \frac{1}{\gamma} \int_{\mathcal{Z}} \mathcal{N}(z';\mu',\sigma')||x - \mu_x(z' * \sigma_p(c) + \mu_p(c))||^2 dz' + \log(2\pi\gamma) \\
= & \frac{1}{\gamma} \int_{\mathcal{Z}} \mathcal{N}(z;\mu_q(x,c),\sigma_q(x,c))||x - \mu_x(z)||^2 dz + \log(2\pi\gamma) \\
= & \mathbb{E}_{q_\phi(z|x,c)}[\log p_\theta(x|z,c)]
\end{aligned}
\tag{33}
$$

Besides, we also have

$$
\begin{aligned}
& \mathbb{KL}[q'(z'|x,c)||\mathcal{N}(0,\mathbf{I})] \\
= & \mathbb{KL}[\mathcal{N}(\mu_q(x,c) - \mu_p(c))/\sigma_p, \sigma_q(x,c)/\sigma_p(c)||\mathcal{N}(0,\mathbf{I})] \\
= & \frac{1}{2}[||\mu_q(x,c) - \mu_p(c))/\sigma_p||^2 + \mathrm{tr}(\sigma_q/\sigma_p) - \kappa - \log(\sigma_q/\sigma_p)] \\
= & \frac{1}{2}[\log\sigma_p - \log\sigma_q - \kappa + ||\mu_q - \mu_p||^2/\sigma_p + \mathrm{tr}(\sigma_q/\sigma_p)] \\
= & \mathbb{KL}[q_\phi(z|x,c)||p_\theta(z|c)]
\end{aligned}
\tag{34}
$$

In terms of generation equivalence, for any $z_p' \sim \mathcal{N}(0, \mathrm{I})$, we have

$$
\begin{aligned}
& p'(x|z', c)p(z'; 0, \mathrm{I}) \\
=& p_\theta(x|z' * \sigma_p(c) + \mu_p(c))p(z'; 0, \mathrm{I}) \\
=& p_\theta(x|z)p(z; \mu_p(c), \sigma_p(c)) \\
=& p_\theta(x|z)p_\theta(z|c)
\end{aligned}
\tag{35}
$$

Therefore we conclude that $M'$ and $M$ share the same cost value, i.e. $\mathcal{L}_c(M'; \theta, \phi) = \mathcal{L}_c(M; \theta, \phi)$, and equivalent data generation distributions.