# OpenReview forum: "Learning Manifold Dimensions with Conditional Variational Autoencoders"
_NeurIPS.cc/2022/Conference — NeurIPS 2022 Accept_

### Official Review · Reviewer_pxuR · 2022-07-06

**Rating:** 5
**Confidence:** 5
**Soundness:** 4 excellent
**Presentation:** 3 good
**Contribution:** 3 good

**Summary:**

This paper aims to understand the relationship between the true intrinsic dimension $r$ of the data, which is assumed to be smaller than the ambient space dimension $d$ on which the data lives (i.e. the paper assumes the manifold hypothesis holds, $r < d$), and the latent variable dimension $\kappa$ in Gaussian VAEs. The authors aim to prove that, when $\kappa \geq r$, the optimal ELBO achieved by a Gaussian VAE with fixed decoder variance $\gamma$ behaves like $(d-r)\log \gamma$, and that the number of activate latent dimensions (i.e. those whose posterior variance goes to 0) is equal to $r$. The authors argue that this insight can be used to estimate the intrinsic dimension $r$. The authors then aim to extend this result to the setting of conditional VAEs (CVAEs). In a final result, the authors aim to show that the practice of sharing weights between the prior and the approximate posterior, which is commonly used in CVAEs, cannot achieve the same optimal ELBO value as could be achieved without the parameter sharing from their earlier result. The authors then claim that this means the practice of sharing these parameters should not be used.

Unfortunately, the paper is very poorly written, to the point where even mathematical definitions cannot be understood in a precise manner. While I detail why I think this is the case below, I highlight that I am giving this paper a soundness score of 2 not because I necessarily think the results are wrong, but presentation is so unclear that it's very hard to judge. Similarly, I give this a contribution score of 2 not because I think the contribution itself would be bad (I actually think a properly re-written version of the paper could be quite strong), but because once again, it's very hard to judge. My confidence score of 5 aims to reflect the fact that I feel very confident that the paper has very poor exposition.

-------------------------------------------------
UPDATE AFTER REBUTTAL
-------------------------------------------------

The authors have significantly addressed the clarity issues present in their original submission. While I still believe the contribution over [4] on the active dimensions being used at optimality is somewhat oversold, this is not the only contribution made by the paper, and I believe the insights presented about conditioning and intrinsic dimension will be valuable to the community. I have thus increased my score accordingly.

**Questions:**

I have the following questions:

16. It is claimed that "this has never been rigorously proven" in the abstract, and that "this has only been proven under an assumption that the decoder is linear or approximately linear [2,4]" when talking about the number of active dimensions estimating the intrinsic dimension. Could you please explain why this is the case? I do not think [4] uses neither a linear nor an approximately linear assumption, and it is also not clear how exactly the results in [4] (particularly the discussion after equation 9 of the arxiv version of [4]) are not rigorous.

17. In Table 4, it is not clear how the AD values are obtained: were these averaged over different $c$'s?

18. How relevant is the VAE setup here? For example, [B] obtains very similar results about the value of the optimal log-likelihood to those of Theorem 1, in a general context of density estimators rather than just VAEs. Similarly, [C] extends the result of [A], which is quite related to the results in the paper being reviewed, from VAEs to density estimators in general. It feels like the ideas presented here about conditional models might be extended to the general setting of density estimators as well. Could you comment on this? To be clear, I am aware that [B] and [C] are very recent and I do not see them not being discussed or compared against in the submitted version as a problem, although I do believe some discussion should be added when the paper is updated.


[B] LIDL: Local Intrinsic Dimension Estimation Using Approximate Likelihood; Tempczyk, Michaluk, Garncarek, Spurek, Tabor & Golinski (2022)

[C] Diagnosing and Fixing Manifold Overfitting in Deep Generative Models; Loaiza-Ganem, Ross, Cresswell & Caterini (2022)

**Limitations:**

I find it hard to properly assess the limitations of this work in its current state given its poor presentation. If my doubts are clarified during the review process, particularly point 8 about definition 4, and question 16 about the differences with [A], I will update this section later on. That being said, part of the point of this paper is that intrinsic dimension can be estimated with VAEs, yet no comparison is carried out against other methods of estimating intrinsic dimension, e.g. [D].

[D] Maximum Likelihood Estimation of Intrinsic Dimension; Levina & Bickel (2004)

**Strengths And Weaknesses:**

## Strengths

I believe this paper considers a relevant question in VAEs, whose answer will be of interest to the community. I also believe that the explicit consideration of the interaction between conditioning and how this affects intrinsic dimension to not only be relevant, but also understudied. The authors have correctly spotted a poorly understood area, and I think this type of research to be valuable, when properly communicated.

## Weaknesses

As I mentioned in my summary, I believe the biggest weakness of this paper is poor exposition. Below I give a list of points that are not properly explained in the mathematical exposition paper (in order of appearance):

1. In definition 1, $\gamma$ is called a "tunable scalar", this really makes it unclear whether or not it is a parameter of the VAE or not. It only becomes clear later, after continuing to read the paper, that the authors mean a fixed hyperparameter.

2. Again, in definition 1, functions being assumed to be "sufficiently complex" is not a precise mathematical assumption and should be dropped from formal definitions.

3. In definition 2, $\gamma_0$ is assumed to be "arbitrarily small" satisfying $\gamma_0 \leq \gamma$. This is again highly confusing, as $\gamma$ was previously "defined" as being part of the definition of a $\kappa$-simple VAE, so this phrasing makes it seem like the VAE does not depend on $\gamma_0$. Yet, the requirement that $\sigma_z(x)^2_j = O(f(\gamma_0))$ would be nonsensical if $\sigma_z(x)^2_j$ did not depend on $\gamma_0$. In other words, either $\sigma_z(x)^2_j$ depends on $\gamma_0$, in which case we just have a $\kappa$-simple VAE with $\gamma_0$ and the requirement that $\gamma_0 \leq \gamma$ makes no sense; or $\sigma_z(x)^2_j$ does not depend on $\gamma_0$, in which case it's $\sigma_z(x)^2_j = O(f(\gamma_0))$ that doesn't make sense. The notation from this paper is clearly inspired by that of [A], and I would recommend the authors take this one step further and make the dependence of the involved neural networks on their parameters more explicit, as in [A]. For example, changing $\sigma_z(x)^2$ to $\sigma_z(x; \phi)^2$ would make statements much more mathematically precise, e.g. $\sigma_z(x; \phi^*_{\gamma_0})^2_j = O(f(\gamma_0))$ is much clearer.

4. Again, there is yet more ambiguity in definition 2: the phrasing "where $f$ is any function that goes to $0$ when $\gamma_0$ goes to zero" does not make sense. Requiring that $\sigma_z(x)^2_j = O(f(\gamma_0))$ for any such $f$ is equivalent to requiring that $\sigma_z(x)^2_j = 0$. I believe the authors meant that there exists one such $f$, not that this holds for any $f$. Furthermore, even if I am correct and the authors meant that such an $f$ exists, I believe this would simply be equivalent to requiring that $\sigma_z(x; \phi^*_{\gamma_0})^2_j \rightarrow 0$ as $\gamma_0 \rightarrow 0^+$, which would be a much clearer way to state the requirement (big-O notation is really meant to bound convergence rates, which do not seem to be of relevance in the definition if $f$ is merely required to exist).

5. Theorem 1 says "the global optimal...", which implicitly assumes the solution is unique in $\theta$ and $\phi$, which I do not believe holds if the neural networks involved are "sufficiently complex".

6. Statement $(i)$ in Theorem one provides $\int_\mathcal{X} (d-r) \log \gamma_0 + O(1) \omega_{gt}(dx)$, an integral which does not seem to depend on $x$, and which I believe can be written simply as $ (d-r) \log \gamma_0 + O(1) $. It would also be good to be more precise as to what is considered a constant and what is not in the $O(1)$ notation. Finally, similarly to the point above, it is not clear what is meant by "can be uniquely achieved", and I do not believe this holds: for example, if a particular VAE configuration achieves the optimal ELBO, applying an orthogonal transformation to the input of the decoder would leave $p(x|z)$ unchanged, and thus achieve the exact same optimum using a different function (and changing the encoder appropriately).

7. In definition 3, the prior is missing from the definition of $\kappa$-simple CVAE. While this is not relevant for VAEs as the prior is just assumed to always be an isotropic Gaussian, this is relevant to the ensuing discussion about the prior being learned as well in the context of CVAEs.

8. Definition 4: This is where it really gets incomprehensible. What is meant by subindexing a vector with $t$? That is, $\varphi(x)$ is, by definition, a vector in $\mathbb{R}^r$. In the manuscript, subindexing with an integer has been used before to denote coordinates (e.g. $\sigma_z(x)^2_j$ is the $j^{th}$ coordinate of $\sigma_z(x)^2$, which by the way is actually defined as a matrix in definition 1, but at least this abuse is understandable), so it seems that $\varphi(x)_t$ should be the $t^{th}$ coordinate of $\varphi(x)$, but the authors go on to say that $\varphi(x)_t \in \mathbb{R}^t$ without explaining exactly what $\varphi(x)_t$ means. Is it the first $t$ coordinates of $\varphi(x)$? Furthermore, even if the previous questions were answered, what is $x$ here? The definition involves $g(c) = \varphi(x)_t$, but it is never explained what $x$ is: the left side seems to depend on $c$ but not $x$, and the right side on $x$ but not $c$.

9. Definition 5: Same issues as in definition 2.

10. Theorem 2: Same issues as in Theorem 1.

11. Corollary 2.1: $\theta^*$ and $\phi^*$ are defined as optimal solutions to a CVAE, but given the loss defined for CVAEs in equation 2, these optimal solutions should depend on $c$, and they do not. Usually the objective being minimized for CVAEs is not really equation 2, but rather an expectation of equation 2 over some prior $p(c)$, so it is not clear exactly what $\theta^*$ and $\phi^*$ mean, as they do not seem to depend on $c$.

12. Theorem 3: Once again, the notation for the loss on equation 3 is sloppy and not too clear, as the actual loss should not really depend on $l$, but the RHS does. Is there a missing summation over $l$ somewhere? Or should $x_l$ be replaced by $x_{\geq l}$?

Now I move on to other, non-mathematical, issues I have with the presentation of the paper:

13. The authors write down some issues, (i) through (iv) in the introduction and keep referring to them, and the readers keeps having to go back to see what the point was. It would be clearer to avoid this.

14. In line 161, the prior in CVAEs is defined as $p_\theta(z|c)$, which is rather confusing notation as $\theta$ is already used to denote the parameters of the decoder. This notation obfuscates the discussion about parameter sharing, and becomes even more confusing when using $q_\theta(z|x,c)$ in line 162.

15. Several experiments are carried out on simulated data, but not enough details are given about how this was generated. For example, section 4.3 should really refer to a detailed appendix explaining the setup in more detail.

Finally, the paper has several grammatical errors, typos, and minor points which are technically incorrect; all of which I would encourage the authors to try to fix. While I see these points as very minor compared to the other issues, I list some below:

- line 32: "each could like on a manifold"
- line 33: "manifold dimension of an image of 1 ...": a single image has intrinsic dimension 0, it is the manifold itself which has intrinsic dimension $>0$.
- lines 35-36: "Such an example ... as the conditioning variable" is pretty hard to understand as a sentence.
- line 128: "Theorem.1"
- line 182: "The cost of VAE balances the three modules with the two terms" is also hard to understand.
- Citations are also sloppy: [4] and [5] are the same, and at least one paper is cited as an arxiv preprint rather than as a conference paper, namely [13], which appeared in ICLR 2021. Note that [13] is simply the one I noticed, but I did not extensively check this, and would not be surprised if this happened more than once.


[A] Diagnosing and Enhancing VAE models; Dai & Wipf (2019)

---

> ### Author Response · Authors · 2022-08-02
> **Response to Reviewer pxuR**
>
> We thank the reviewer for the detailed review as well as the suggestions for improvement. Also, we appreciate that the reviewer recognize the question our paper studies is relevent and understudied.
>
> Based on the review, we have uploaded a new pdf with updated definition and theorem statements. Some text descriptions are also revised. Hope this may provide a clearer presentation of our work.
>
> Next, we answer the questions in the review:
>
> **Answers to Weaknesses**
>
> Thanks for the detailed review. Apart from the below answers, we have addressed many points in the updated version.
>
> 1. $\gamma$ is a trainable scalar in $\theta$.
> 2. We have made it more specific with "$\mu_z(x)$ and $\mu_x(z)$ are arbitrarily $L$-Lipschitz continuous".
> 3. We have revised Definition 2 accordingly (Definition 3 in our new version).
> 4. We have revised Definition 2 accordingly (Definition 3 in our new version).
> 5. Yes, the optimal solution is not unique. Thank you for pointing it out. We replace it with "any global optimal solutions".
> 6. We have revised Theorem 1 and 2.
> 7. We have added prior to both VAE and CVAE definitions.
> 8. We have updated the notation in the definition of *effective dimension*. We also illustrate the $\{x, c\}$ pairs in Line 110 to make it more clear.
> 9. We have fixed them according to the reviews.
> 10. We have fixed them according to the reviews.
> 11. In the updated version, we consider each $x \in \mathcal{X}$ has a corresponding $c$. In this sense, an integral over $\mathcal{X}$ would also cancel $c$ on the right hand side. We will consider to improve the paper with a more precise formulation in its finalized version.
> 12. Fixed by summing over $l$.
> 13. Thank you for the suggestion. We will consider this point and improve the paper for its finalized version.
> 14. For $p_\theta(z| c)$ we add a footnote that that we slightly abused the notation of $\theta$ as parameters for both the prior and decoder. For $q_\theta(z|x, c)$, it is a typo, and should be $q_\phi(z|x, c)$.
> 15. We have added some details of the experiments both in the captions and texts.
> 16. Thank you for catching the typos and grammatical errors. We have fixed them in the updated version.
>
> **Answers to Questions**
>
> - It is claimed that "this has never been rigorously proven" in the abstract, and that "this has only been proven under an assumption that the decoder is linear or approximately linear [2,4]" when talking about the number of active dimensions estimating the intrinsic dimension. Could you please explain why this is the case?
>
> Just to clarify, reference [2] proves that active dimensions at VAE global minima will estimate the correct/true principal subspace of the data under the assumption of a linear decoder.  However, this work does not consider nonlinear manifolds. In contrast, reference [4] does consider general nonlinear manifolds, but does not rigorously prove that when such manifolds are present, and the decoder is nonlinear and suitably matched as well, that the number of learned active dimensions of VAE global minima will align with the true data manifold dimension.
>
> - In Table 4, it is not clear how the AD values are obtained: were these averaged over different $c$'s?
>
> Yes, they are averaged over different classes.
>
> - How relevant is the VAE setup here? For example, [B] obtains very similar results about the value of the optimal log-likelihood to those of Theorem 1, in a general context of density estimators rather than just VAEs. Similarly, [C] extends the result of [A], which is quite related to the results in the paper being reviewed, from VAEs to density estimators in general. It feels like the ideas presented here about conditional models might be extended to the general setting of density estimators as well. Could you comment on this? To be clear, I am aware that [B] and [C] are very recent and I do not see them not being discussed or compared against in the submitted version as a problem, although I do believe some discussion should be added when the paper is updated.
>
> Thanks, these are very interesting references and we will add them to our related works. However, our paper is about understanding the properties of \(C\)VAE and its ability for estimating the ground truth manifold dimensions. If other techniques are capable of doing it, that is interesting but separately from our purpose.

---

> > ### Comment · Reviewer_pxuR · 2022-08-03
> > **Additional questions**
> >
> > I thank the authors for their reply. I have read the updated version of the manuscript and think that the mathematical presentation is much improved. I have some follow-ups, however:
> >
> > 1. Definition 5 is still poorly phrased. The way it is currently phrased implies that for every $x$, there exists a $g$ such that $g(c)=\varphi(x)_t$, which I do not think is the intended definition, as this would always be trivially satisfied. I believe what is meant is that there should exist a single $g$ such that for every $x$, $g(c)=\varphi(x)_t$.
> >
> > 2. I am still not convinced about some differences with [4]. You say that `Just to clarify, reference [2] proves that active dimensions at VAE global minima will estimate the correct/true principal subspace of the data under the assumption of a linear decoder. However, this work does not consider nonlinear manifolds. In contrast, reference [4] does consider general nonlinear manifolds, but does not rigorously prove that when such manifolds are present, and the decoder is nonlinear and suitably matched as well, that the number of learned active dimensions of VAE global minima will align with the true data manifold dimension.` I agree about [2], but that is not really what I meant, and the difference with [4] is still unclear to me: Theorem 5 in [4] shows that, as long as $\kappa \geq r$, VAEs can achieve perfect reconstructions. From there, it seems straightforward to argue that only $r$ dimensions will be active, and that the remaining $\kappa - r$ will have their variance set to $1$ so as to minimize their KL against the prior. Indeed, this argument is made in [4] `Therefore, in the neighborhood of optimal solutions the VAE will naturally seek to produce perfect reconstructions using the fewest number of clean, low-noise latent dimensions`, which although not a formal statement in the form of a theorem, seems to say exactly the same thing being said here as a direct consequence of Theorem 5 in [4]. Could the authors please further clarify?

---

> > > ### Author Response · Authors · 2022-08-06
> > > **Follow-up Response for Reviewer pxuR**
> > >
> > > We appreciate your quick follow ups, and hope our answers below have addressed them:
> > >
> > > - Definition 5 is still poorly phrased. The way it is currently phrased implies that for every $x$, there exists a $g$ such that $g(c)=\varphi(x)_t$, which I do not think is the intended definition, as this would always be trivially satisfied. I believe what is meant is that there should exist a single $g$ such that for every $x$, $g(c)=\varphi(x)_t$.
> > >
> > > **A**: Thanks for pointing out the confusion, and we'll update the main text accordingly. What we really try to say is that for every $x$ and its condition $c$,
> > >
> > > 1. There exists a **diffeomorphism** $g$ and an integer $t$ such that $g(c)=\varphi(x)_t$, and
> > > 2. There is no **diffeomorphism** $g$ satisfying  $g(c)=\varphi(x)_{t+1}$.
> > >
> > > Consider this case as a visualization of definition 5: for any $x\in \mathcal{X}$, we let its $c$ be the first dimensions of $\varphi(x)$. Obviously, let $g$ be identity mapping and $t=1$, we have $g(c)=\varphi(x)_t$, and there exists no diffeomorphism $g$ such that $g(c)=\varphi(x)_2$. In this case the effective dimension is 1.
> > >
> > > Essentially effective dimension characterizes the maximum number of dimensions of the manifold for $x$ that $c$ can recover. With $t$ being the effective dimension, we agree that it might be trivial to find a desired $g$, but at the same time it is impossible to have such a $g$ if $t$ goes larger than that. In definition 5, we use the existence of such a $g$ as a tool to introduce the concept of effective dimension, rather than examine possible examples of $g$s.
> > >
> > > - Could the authors please further clarify?
> > >
> > > **A**: Theorem 5 in reference [4] only shows that perfect reconstructions will occur, but it does not at all suggest that this perfect reconstruction will be achieved at global optima with the fewest number of active dimensions.  And it is easy to envision underdeterimined scenarios whereby there exists a multitidude of solutions capable of producing zero reconstruction error with varying numbers of active dimensions (this is analogous to underdeterimined linear systems with an infinite number of feasible solutions with varying $L_0$ norm). While it is true that later, below Theorem 5 in [4] there is some loose reasoning to suggest that unnecessary latent dimensions might be pruned away by the VAE, there is definitely not anything close to a formal proof.  In fact this was part of our original motivation, to formally prove the conjecture first advanced in [4].

---

> > > > ### Comment · Reviewer_pxuR · 2022-08-08
> > > > **Reply**
> > > >
> > > > I thank the authors for their reply. My main concerns about clarity have been adequately addressed and I am increasing my score, as I believe the mathematical presentation is now clear and understandable. I would just like to ask the authors to appropriately modify the camera-ready version to include not only the clarity changes that have mostly already been made, but also the discussion about Theorem 5 in [4].

---

> > > > > ### Author Response · Authors · 2022-08-09
> > > > > **Response to Reviewer pxuR**
> > > > >
> > > > > Thanks for your continued engagement with our paper. And per the reviewer's suggestion, we can certainly update the paper to include all the discussed changes, noting that NeurIPS allows for an additional (10th) page to address reviewer comments in the final version.

---

### Official Review · Reviewer_ZAHe · 2022-07-08

**Rating:** 6
**Confidence:** 4
**Soundness:** 4 excellent
**Presentation:** 2 fair
**Contribution:** 3 good

**Summary:**

This paper studies the behaviour of variational auto-encoders (VAEs) and conditional variational auto-encoders (CVAEs) in the presence of data supported on a low-dimensional manifold.
The authors demonstrate that VAEs are able to learn the intrinsic manifold dimension of the data at optimality.
They also show that a similar result exists for CVAEs, and that effective conditioning should reduce the loss function at optimality.

The authors then probe some common design choices in VAEs and CVAEs, noting the following:
1. Conditioned and unconditioned priors are theoretically equivalent.
2. Learning the decoder variance should result in better performance.
3. A common weight-sharing technique in autoregressive models should be avoided.

The experiments section then directly addresses their claims, particularly on the synthetic dataset with known groundtruth intrinsic manifold dimension.

### EDIT AFTER REBUTTAL

I am happy with the discussion that we've had and believe the clarity of the paper can be easily increased beyond the recent revisions which have themselves already increased the clarity. Thus, I am raising my score to a *Weak Accept*.

**Questions:**

These are the main questions I have of the authors. There are others listed above as well but they are more minor.
1. Can the authors provide an explanation of the link between active latent dimensions and some notion of intrinsic dimensionality of the generative model in data space?
2. How do we reconcile the simple Riemannian manifold assumption with the idea of using conditioning to learn "unions of manifolds"?
3. How can we expect to learn $\gamma$ at all in light of the discussion in Section 3.2?
4. How valid is maximizing likelihood in the presence of manifold-supported data?
5. How can we say that the results on MNIST and Fashion MNIST are consistent with intuitive visual complexity?
6. Can we get $t$ in practice for realistic conditioning variables, and if so can we get more practical understanding of intrinsic dimension from this?

**Limitations:**

I think the authors have done a reasonable job of addressing the practical limitations of their work.

**Strengths And Weaknesses:**


## Overall Assessment

Along the standard review dimensions of __quality__, __clarity__, and __significance__, we have *both* strengths and weaknesses, while on the dimension of __originality__ we have only strengths.
Altogether, I believe the weaknesses outweigh the strengths to some degree, and thus I would weakly recommend rejection at this time.
If the authors can adequately answer my questions about the paper overall and perhaps address some of the issues with __clarity__, I would be inclined to increase my recommendation.
I'll elaborate below.

## Strengths

1. I consider part (ii) of __Theorem 1__ to be a major strength of the paper. The utility of this result is clear: VAEs which are well-optimized have the ability to estimate intrinsic manifold dimension, under the assumption that we have a reliable method to estimate the number of active latent dimensions.
2. I found the paper overall to be a fairly original work. Of course there are quite strong connections to previous work -- namely the _Diagnosing and Enhancing VAEs_ paper -- but the analysis provided here is sufficiently novel and comes with further useful insights.
3. I found the introduction and motivation of the paper to be very clear, and the general storyline to be straightforward to follow.
4. __Theorem 2__ part (i) is quite sensible and intuitive: adding good conditioning variables can reduce the loss function.
5. I find Section 3.1 to be quite interesting.
6. I think pursuing ideas around "unions of manifolds" to be quite intuitive and worthwhile. I appreciate that this paper is bringing this idea to the table.
7. __Theorem 3__ is also an interesting result that appears to shed light on an issue with a common practical technique.
8. Table 1 is a strong result - it is nice to see that $r$ = AD in almost all cases where $\kappa \geq r$.
9. Table 3 is also a strength - we can clearly see AD = $r - t$ here.
10. Table 8 is a strong result.

## Weaknesses

There are some major weaknesses to which I would first like to devote individual sections, and then I'll comment on minor issues later on.

### On the Relationship Between Active Latent Dimensions and Manifold Dimension

Theorem 1 shows that a VAE trained to optimality will have its number of active latent dimensions exactly equal to the intrinsic manifold dimension of the data.
We would ideally also have some result that directly relates the number of active latent dimensions to the intrinsic dimensionality of the overall generative model, so that we could say samples from the generative model have dimensionality approximately equal to the true data dimension.
Yet I cannot easily see such a link between the two quantities, which reduces the significance of the result.
It is stated in the text that active latent dimensions are those "used for reconstruction"; is it then the case that the "inactive" latent dimensions do not provide any dimensionality to the generative model?

Admittedly, it is possible that I have not fully understood Definition 2 in-depth, but I would also argue that the Definition 2 (and Definition 5 by extension) is itself not very clear.
I am left lacking intuition on what active dimensions actually _mean_: simply stating that the encoder variance goes to zero with $\gamma$ does not provide much insight (and by the way, how does $\gamma_0$ factor in here? It appears that we are replacing $\gamma$ in the $\kappa$-simple VAE definition with $\gamma_0$, but then why do we need to say $\gamma_0 \leq \gamma$?).

Some clarity and further discussion would be appreciated around both of these points here.

### Significance of Theorem 2 / Corollary 2.1 and CVAE Analysis

As far as I understand, Theorem 2 part (ii) and Corollary 2.1 (by the way, I think there is an error in Corollary 2.1: shouldn't the active dimensions be $r-t_1$ and $r-t_2$, respectively?) are statements about the mechanics of the encoder in latent space.
Yet I am again left wondering how this relates to the dimensionality learned by the generative model in data space, as I see no direct correspondence between the two.
Why should changes in effective dimensionality in the latent space correspond to changes in intrinsic dimensionality in the data space?
How does this result in "different data manifolds"?

I am also a bit confused about how the proposed capacity of CVAEs to learn unions of manifolds relates to the original assumption from L61 that states the data lives on a simple Riemannian manifold with constant dimension and a single chart.
How do we reconcile these two seemingly contradictory points?


### Minor Weakness, Issues, and Calls for Clarification

1. I appreciate that Eq. (1) is still a lower bound on the average log-likelihood of the density $p_\theta$ over the data distribution $\omega_{gt}$. However, I am curious about the authors' opinion of the validity of the maximum likelihood objective for a full-dimensional density in the presence of manifold-supported data? In particular, we can no longer interpret the maximum likelihood objective as minimizing the KL divergence from $\omega_{gt}$ to the distribution induced by $p_\theta$ over $\mathbb R^d$, as $\omega_{gt}$ is not absolutely continuous with respect to $p_\theta$'s distribution.
2. Along the same vein, it is inappropriate to compare NLL/ELBO values across data with different manifold dimensions and thus dominating measures; yet the experiment section appears to do this a few times. Some care here would be prudent.
3. I find Section 3.2 difficult to understand. How can we expect $\gamma$ to adaptively balance the two terms in the loss function when the optimal learning behaviour is simply to make $\gamma$ as small as possible? Based on this analysis, I find it remarkable that learning $\gamma$ can succeed at all.
4. The headline of Section 3.3 does not accurately represent the scope of Theorem 3, which strictly relates to conditional autoregressive models. While Theorem 3 is indeed interesting, the section purports to make a more general statement about weight sharing that does not appear.
5. What is meant by "sufficiently complex" in L70?
6. [4] and [5] are the same citation.
7. L97: It is claimed that "the ratio of the norm term in reconstruction and $\gamma$ is constant, i.e. the data dimensions $d$." Why wasn't this ever checked in practice?
8. Parentheses appear to be missing from the integrand in lines 85, 133, and 172
9. What is the role of $\kappa$ in Theorems 1 and 2 beyond simply $\kappa \geq r$? Why doesn't $\kappa$ appear in the bounds?
10. In definition 4, is $\phi$ the same $\phi$ as in L62?
11. Why does the prior variance take $x$ as an argument in definition 5?
12. L153-155 is written in a very unclear way.
13. L219-221 is a very weak and nebulous discussion.
14. In L249 it is stated that $c = u_{1:t}$, but earlier we have $c = h'(u_{1:t}) = G'u_{1:t}$. Which one is correct? Did you just take $G' = I_t$?
15. In L239-240 it is stated that "the number of active dimensions equals to data's intrinsic dimension" and L256-257 it is stated that "This characterization of dataset complexity is consistent with their intuitive visual complexity". I would have to strongly disagree with this: in my opinion, MNIST is "less complex" than Fashion-MNIST, so I don't quite understand why it ends up with lower active dimensions and thus lower intrinsic dimension. Would the authors clarify what was meant here?
16. Why is $\kappa = 90$ so large in the Continuous Condition experiment? Is this a typo? Seems very strange.
17. I generally find the details around Table 5 and Table 6 to be lacking. How many datapoints from each class? How is the reconstruction error for each class, i.e. has the CVAE learned each of the manifolds or just the dimensionalities? Is $t$ considered to be $0$-dimensional in the case of Table 5?
18. What is the correct number of AD in Section 4.4?
19. Is there some gap between the theory (Theorem 3) and the result in Table 7 about the equivalence of $p(z)$ and $p(z \mid c)$? It appears that the conditional prior does a better job in the more difficult case of $\log \gamma = 20$ initially. Is it perhaps easier to learn with the conditional prior?

---

> ### Author Response · Authors · 2022-08-02
> **Reponse to Reviewer ZAHe (Part 1)**
>
> We thank the reviewer for the detailed review as well as the suggestions for improvement. Also, we appreciate the reviewer for recognizing the originality of our work and considering three theorems as strong, intuitive and interesting.
>
> Based on the review, we have uploaded a new pdf with updated definition and theorem statements. Some text descriptions are also revised. Hope this may provide a clearer presentation of our work.
>
> Next, we answer the questions in the review:
>
> - I cannot easily see such a link between the two quantities, which reduces the significance of the result.
>
> **A**: Previous work (e.g., [4]) has demonstrated that global minima of VAE models can achieve zero reconstruction error for all samples lying on the data manifold. However, it was not previously demonstrated in a general setting that this perfect reconstruction was possible using a minimal number of active latent dimensions, and hence, it is conceivable for generated samples involving a larger number of active dimensions to stray from this manifold.  In contrast, to achieve perfect reconstruction using the minimal number of active latent dimensions, as we have done under the stated assumptions, implies that generated samples must also lie on the manifold (the noisy signals from inactive dimensions are blocked by the decoder and therefore cannot produce deviations from the manifold).  Very good/insightful question.
>
>
> - It is stated in the text that active latent dimensions are those "used for reconstruction"; is it then the case that the "inactive" latent dimensions do not provide any dimensionality to the generative model?
>
> **A**: Yes. Inactive dimensions are used to optimize the KL term so they do not provide any dimensionality to minimize the reconstruction error. These inactive dimensions are with a variance that has a positive lower bound even as $\gamma$ goes to 0. When such latent dimensions are greater than $d - r$, no matter how many inactive dimensions there are, the model cannot perfectly reconstruct even a single manifold dimension.
>
>
> - simply stating that the encoder variance goes to zero with $\gamma$ does not provide much insight.
>
> **A**: Active dimensions are dimensions of $z\sim q(z|x)$ where each such dimension has its variance $\sigma_z(x)_j^2$ going to zero if $\gamma$ goes to zero, because a small variance enables smaller reconstruction error. On the contrary, each inactive dimension has its variance $\sigma_z(x)_j^2$ going to prior variance, as these dimensions are not necessary helping reconstruction thus instead used for an optimal KL-divergence with the prior. Please refer to the proof for more details.
>
>
> - Why should changes in effective dimensionality in the latent space correspond to changes in intrinsic dimensionality in the data space?
>
> **A**: Thanks for pointing it out. We have fixed the error in Corollary 2.1 and the definition of effective dimension. The encoder learns the dimensionality in the latent space instead of data space.  "different data manifolds" means for data with multiple conditions, under different conditions, the subset of data may have different manifold dimensions. And the changes in effective dimensionality in the latent space determine which subset of the manifold the latent variable $z$ will learn.
>
> - How do we reconcile these two seemingly contradictory points?
>
> **A**: We add a new definition of data for the training data (see Definition 1). When the data is unconditioned, it is exactly lies on a simple Riemannian manifold while when it is conditioned on some $c$, data will lie on a union of manifolds corresponding to $c$.
>
> **Answers to Minor Weakness, Issues, and Calls for Clarification**
>
> Thanks for the detailed review. Apart from the below answers, we have addressed many points in the updated version.
>
> 1. We agree that it could be problematic to apply maximum likelihood to data lies on a low-dimensional manifold, but this is what everyone is doing because images are on a manifold and people apply likelihood-based generative moldes to this data. Therefore, our work provide a new method of analysis on this scenario.
> 2. Yes, we agree that comparing ELBOs under different manifolds can be problematic. However, here we reported negative ELBO as a conventional reported figure, which is not the central part of our analysis and is not for comparison.
> 3. When $\theta$ and $\phi$ are fixed, there exists an optimal $\gamma$, and that $\gamma$ is proportional to the reconstruction loss. If we fix $\gamma$, when reconstruction error decreases, its weight will be smaller than the optimal value, which prevent the model to optimize further.
> 4. We used sequential data as the representation of Theorem 3.
> 5. We have made it more specific with "$\mu_z(x, c)$ and $\mu_x(z, c)$ are arbitrary $L$-Lipschitz continuous functions".

---

> > ### Comment · Reviewer_ZAHe · 2022-08-03
> > **Response Pt. 1**
> >
> > Thanks for your reply and your modifications to the manuscript. I appreciate the discussion here but still have some follow-up questions and points of confusion:
> >
> > - `Previous work (e.g., [4]) has demonstrated that global minima of VAE models can achieve zero reconstruction error for all samples lying on the data manifold. However, it was not previously demonstrated in a general setting that this perfect reconstruction was possible using a minimal number of active latent dimensions, and hence, it is conceivable for generated samples involving a larger number of active dimensions to stray from this manifold. In contrast, to achieve perfect reconstruction using the minimal number of active latent dimensions, as we have done under the stated assumptions, implies that generated samples must also lie on the manifold (the noisy signals from inactive dimensions are blocked by the decoder and therefore cannot produce deviations from the manifold).` Is it possible to include such a discussion in the paper?
> > - __On Corollary 2.1__: Is the point here that the different data manifolds modelled by $c_1$ and $c_2$ will necessarily be of dimension $r - t_1$ and $r-t_2$, respectively?
> > - Fair enough point about how everybody does modelling of low-dimensional data with high-dimensional methods.
> > - I see in your other comment you write out what the optimal $\gamma$ is. Is it possible to include this expression in the main text? I still find sentences like "Specifically, when a VAE model converges to its optimum, γ will go to zero," in the manuscript to be confusing. Some more discussion on the behaviour of $\gamma$ would be helpful.
> > - `We used sequential data as the representation of Theorem 3.` I still think the section is heavily oversold. "Weight Sharing" is far more general than just the specific type employed in that section, and people might get the wrong impression that Theorem 3 extends beyond just sequential data. Can you please modify the title of Section 3.3?

---

> > > ### Author Response · Authors · 2022-08-06
> > > **Follow-up Response Pt. 1 for Reviewer ZAHe**
> > >
> > > We appreciate your quick follow ups, and hope our answers below have addressed them:
> > >
> > > - Is it possible to include such a discussion in the paper?
> > >
> > > **A**: Thanks for the suggestion. We'll include the above discussion in the finalized paper.
> > >
> > > - Is the point here that the different data manifolds modelled by $c_1$ and $c_2$ will necessarily be of dimension $r-t_1$ and $r-t_2$, respectively?
> > >
> > > **A**: Yes, the $t$ of each $c$ represents the maximum number of dimensions of the manifold for $x$ that $c$ can recover. With a $r$-dimensional manifold, the model conditioned on $c_1$ would only need to recover the remaining $r-t_1$ dimensions with $z$, and likewise for $c_2$.
> > >
> > > - I see in your other comment you write out what the optimal $\gamma$ is. Is it possible to include this expression in the main text? I still find sentences like "Specifically, when a VAE model converges to its optimum, γ will go to zero," in the manuscript to be confusing. Some more discussion on the behaviour of $\gamma$ would be helpful.
> > >
> > > **A**: Thanks for the suggestion. As shown in prior work and this paper, a VAE model with sufficient capacity will be such that as we approach any global minimum, $\gamma \rightarrow 0$. We'll include this discussion in the finalized paper.
> > >
> > > - I still think the section is heavily oversold. "Weight Sharing" is far more general than just the specific type employed in that section, and people might get the wrong impression that Theorem 3 extends beyond just sequential data. Can you please modify the title of Section 3.3?
> > >
> > > **A**: While the weight-sharing issue we raise can impact a broad class of VAE models, we agree with the reviewer that as things presently stand, only the sequential case has been directly addressed. We can reword to reflect this limitation and avoid any confusion.

---

> > > > ### Comment · Reviewer_ZAHe · 2022-08-08
> > > > **Reply Conclusion - Both for here and the other comment**
> > > >
> > > > Thank you for the clarification, both here and in the second part of my response.
> > > >
> > > > I think the discussion that we've had has made things much more clear for me, and this discussion should easily be accommodated in the extra page that will be available if this paper makes it to publication. From my perspective, the paper has already been improved, but I believe with further improvements to *clarity* as discussed, this paper would be elevated to the level of *Weak Accept*. I will trust that the authors will provide these improvements and will thus raise my score accordingly.

---

> > > > > ### Author Response · Authors · 2022-08-09
> > > > > **Reply to Reviewer ZAHe**
> > > > >
> > > > > Thanks for your continued engagement with our paper. And per the reviewer's suggestion, we can certainly update the paper to include the discussed changes in the extra (10th) page that NeurIPS allows in the final version.

---

> ### Author Response · Authors · 2022-08-02
> **Reponse to Reviewer ZAHe (Part 2)**
>
> 6. We have revised it in the updated version.
> 7. It can be seen from $\gamma$'s optimal value, i.e. $\gamma^* = \arg \min_\gamma \mathcal{\tilde{L}(\theta, \phi)} = \frac{L^2}{d} \mathbb{E}_{\varepsilon \sim N(0, I)} [||\sigma_z(x)_{1:r} \varepsilon||^2]$
> 8. Remove integration over data
> 9. $\kappa$ is defined in $\kappa$-simple VAE model definition, i.e. the dimension of latent variable in the model. In the bounds as long as $\kappa \geq r$, $\kappa$ will not affect the rate of the bounds, so it is involved in the constant $O(1)$.
> 10. Yes, there are the same.
> 11. It is a typo. Thanks for pointing it out.
> 12. Consider this example: the conditioning variable equals to two dimensions of the manifold that data lie on. In this case, the conditioning variable provides information for the two dimensions thus the number of active dimensions would be two less. We will improve the statement in the finalized paper.
> 13. We intended to describe that the two datasets are well balanced between simple synthetic data and high-resolution natural images. For synthetic data, the mapping between the manifold and ambient space might be too simple to fit by neural networks. On the other hand, high-resolution natural images may require non-trivial efforts in architecture design to capture the mapping, which is not the main focus of our paper. We will improve the statement in the finalized paper.
> 14. Yes, here we use identity mapping, and we have clarified it in the updated text.
> 15. Yes. The number of dimensions depends on two things: one is the complexity of the dataset, the other is the structure of the decoder. Thus it is a function of the capacity of the decoder, and the decoder could be better aligned with one dataset but not the others.
> 16. A tuned hyper-parameter, for better performance. Note that $\kappa$ can be arbitrarily large without affecting the model performance, since an optimal model only has $r$ active dimensions.
> 17. The ELBO term shows we have learnt the manifolds. For the problem of $t=0$, we need to consider the data. When data is a union of manifolds, without the condition, the model need to use the maximal intrinsic dimension among the manifolds for reconstruction, which will result in an increase in the loss. On the other hand, if we add a discrete $c$ to indicate the manifold, the difference of the number of active dimensions is $t$.
> 18. Thanks for pointing it out. The parameters of data are now listed in the caption.
> 19. It is to show an extremely large initial $\gamma$ will have a negative impact on the stochastic optimization of training, and that leads to a compromised model performance.
>
>
> **Answers to Questions**
>
> - Can the authors provide an explanation of the link between active latent dimensions and some notion of intrinsic dimensionality of the generative model in data space?
>
> Yes. When the decoder is a L-Lipschitz continuous function and learns the ground truth diffeomorphism, the number of active dimensions determines intrinsic dimensionality of the generative model in data space.
>
> - How do we reconcile the simple Riemannian manifold assumption with the idea of using conditioning to learn "unions of manifolds"?
>
> We have updated the paper, with more precised definition and description.
>
> - How can we expect to learn $\gamma$ at all in light of the discussion in Section 3.2?
>
> In point 3. When $\theta$ and $\phi$ are fixed, there exists an optimal $\gamma$, and that $\gamma$ is proportional to the reconstruction loss. If we fix $\gamma$, when reconstruction error decreases, its weight will be smaller than the optimal value, which prevent the model to optimize further.
>
> - How valid is maximizing likelihood in the presence of manifold-supported data?
>
> In point 1. This is a really valid question. We agree that it could be problematic to apply maximum likelihood to data lies on a low-dimensional manifold, but this is what everyone is doing because images are on a manifold and people apply likelihood-based generative moldes to this data. Therefore, our work provide a new method of analysis on this scenario.
>
> - How can we say that the results on MNIST and Fashion MNIST are consistent with intuitive visual complexity?
>
> On the one hand, the number of dimensions depends on two things: one is the complexity of the dataset, the other is the structure of the decoder. Thus it is a function of the capacity of the decoder, and the decoder could be better aligned with one dataset but not the others. On the other hand, in some way Fashion MNIST has been normalized and centered more clearly than MNIST, so we can reasonably expect the active dimensions of MNIST is larger.
>
> - Can we get $t$ in practice for realistic conditioning variables, and if so can we get more practical understanding of intrinsic dimension from this?
>
> Yes we can. First train a vanilla VAE to get the intrinsic manifold dimension $r$ of the data, then train a CVAE to get the conditioned manifold dimension $r-t$.

---

> > ### Comment · Reviewer_ZAHe · 2022-08-03
> > **Response Pt. 2**
> >
> > - `A tuned hyper-parameter, for better performance` I appreciate that the theory says you can make this as large as you want, but it still feels very strange! I think it is worth considering or discussing why a latent dimension of $90$ would be useful in such a setting, as the intrinsic dimensionality is only $12$ and the ambient dimension is only $20$. I don't think this should simply be dismissed.
> > - `It is to show an extremely large initial  will have a negative impact on the stochastic optimization of training, and that leads to a compromised model performance.` Yes I am aware of the intention of the experiment, but my question still stands: the conditional prior seems to do better than the unconditional prior in the difficult case of $\log \gamma = 20$. I think this is worth delving into deeper, and may indicate a disparity between the theory that unconditional priors are equivalent to conditional priors, and the practical result that conditional priors might end up working better in some cases.
> > - `Yes. When the decoder is a L-Lipschitz continuous function and learns the ground truth diffeomorphism, the number of active dimensions determines intrinsic dimensionality of the generative model in data space.` as with my other reply, it may be worth stating this more plainly in the text?

---

> > > ### Author Response · Authors · 2022-08-06
> > > **Follow-up Response Pt. 2 for Reviewer ZAHe**
> > >
> > > - I appreciate that the theory says you can make this as large as you want, but it still feels very strange! I think it is worth considering or discussing why a latent dimension of 90 would be useful in such a setting, as the intrinsic dimensionality is only 12 and the ambient dimension is only 20.
> > >
> > > **A**: Decreasing the latent dimension in practice reduces the decoder capacity, thus in order to achieve on par performance we need to tweak the architecture and model dimensions in other layers. After adding more layers to the decoder, we observe that when $\kappa=20$, the model can learn the correct number of active dimensions with small reconstruction error. The following is a table with $r = 12, d = 20, \kappa = 20$. We'll include this analysis and result in the finalized paper.
> > >
> > > | $t$ | True AD | AD with attention |  Recon | -ELBO with attention |
> > > |:---:|:-------:|:-----------------:|:------:|:--------------------:|
> > > |  2  |    10   |         10        | 0.0066 |        -41.487       |
> > > |  4  |    8    |         8         | 0.0146 |        -20.522       |
> > > |  6  |    6    |         6         | 0.0051 |        -73.259       |
> > > |  8  |    4    |         4         | 0.0057 |        -80.642       |
> > > |  10 |    2    |         2         | 0.0141 |        -55.140       |
> > >
> > > ------
> > >
> > > - Yes I am aware of the intention of the experiment, but my question still stands: the conditional prior seems to do better than the unconditional prior in the difficult case of $\log \gamma = 20$. I think this is worth delving into deeper, and may indicate a disparity between the theory that unconditional priors are equivalent to conditional priors, and the practical result that conditional priors might end up working better in some cases.
> > >
> > > **A**: Thanks for diving deeper in that. In short, the reasons for a gap at $\log \gamma = 20$ are:
> > >
> > > 1. Models with a bad initialization can potentially converge to sub-optimal solutions (note that our theory only applies to globally-optimal solutions).
> > > 2. For the unconditional prior in numerical experiments, We did not construct the corresponding decoder in such a way that it was strictly equivalent to the model with a conditional prior.
> > >
> > > In the proof of Remark 1 we have shown that the equivalence between a trained CVAE model with a parameterized prior and an unparameterized one, and this equivalence is strictly achieved by moving existing computational components in the parameterized one without any further training.
> > >
> > > We prove the equivalence by construction, thus if we build the unparameterized CVAE from any parameterized one in numerical experiments, we should observe identical training loss and inference results.
> > >
> > > On the other hand, the proof shows the existence of equivalence without excluding the other possible ones. In the numerical experiments, we intend to show that in practice, given enough model capacity and good training parameters, one can directly train an unparameterized CVAE and achieve similar performance, instead of starting from training a parameterized one. Having $\log \gamma = 20$ means the models are not optimally initialized and that would impact model convergence negatively.
> > >
> > > The following is the result from a CVAE with an unparameterized prior and a decoder with increased capacity, where $d = 20, r = 10, \kappa = 20, t = 5$. Together with Table 7 in the paper, we can conclude that in practice the performance at convergence heavily depends on both model architecture design and hyperparameters. Results with less careful architecture or hyper-parameters may not reflect the true global optimal.
> > >
> > > | Init $\log \gamma$ | AD |  -ELBO  |
> > > |:------------------:|:--:|:-------:|
> > > |         -20        |  5 | -41.202 |
> > > |         -10        |  5 | -44.530 |
> > > |          0         |  5 | -44.375 |
> > > |         10         |  5 | -43.717 |
> > > |         20         |  5 | -45.217 |
> > >
> > > We will include the above details to motivate this experiment and emphasize its differences with Remark 1 in the finalized paper.
> > >
> > >
> > > - As with my other reply, it may be worth stating this more plainly in the text?
> > >
> > > **A**: Thanks for the suggestion.
> > >
> > > "When the decoder is an L-Lipschitz continuous function and learns the ground truth diffeomorphism, the number of active dimensions determines intrinsic dimensionality of the generative model in data space."
> > >
> > > We'll include this statement in Section 2.1, following Theorem 1, in the finalized paper.

---

### Official Review · Reviewer_R6JP · 2022-07-10

**Rating:** 6
**Confidence:** 3
**Soundness:** 2 fair
**Presentation:** 2 fair
**Contribution:** 3 good

**Summary:**

In this work, the authors theoretically analyze the properties of Variational Autoencoders (VAEs) and Conditional VAEs (CVAEs) when the data lies on a low-dimensional manifold. Notably, they demonstrate, under assumptions, VAEs will converge to a solution that will correspond to using the same number of latent dimensions (active dimensions) as the number of dimensions in the data manifold; therefore, allowing for the identification of the number of manifold dimensions. They also extend the analysis when the data lies on a mixture of low-dimensional manifolds. Further, the authors analyze several active practices in their framework and provide theoretical insights into why they may or may not work. Finally, the authors offer some experimental support to their claims.

**Questions:**

Please refer to the Strengths and Weaknesses section.

**Limitations:**

I think the authors captured the limitation reasonably well in their last comment in the paper. I agree with the authors and believe their insights must be tested thoroughly in different empirical settings. As is, the take-aways from the paper seem minimal if the experiments do not follow the results. Please refer to the Strength and Weaknesses section for more detailed comments.

**Strengths And Weaknesses:**

# Strengths
### Theoretical Analysis.
In this paper, the authors propose theoretical insights into the behavior of VAE and Conditional VAE when the data lies on low dimensional manifolds. Under assumptions, the converged solutions can also be used to determine the manifold dimensions. The authors also analyze several active practices in their framework and provide theoretical insights into why some of them may or may not work. I think there are several interesting and important theoretical insights in this paper.


# Weakness
### Theorem 1
I am unclear about what it means to find an optimal solution under $\gamma = \gamma_0$. From definition 1, it seems that $\kappa$-simple VAE is defined under the assumption of $\gamma$ being a tunable parameter. Can the authors comment on what I am missing here?

### Definition 4
What is $\psi(x)$ here? Is it the same as L52? Are these the first $t$ dimensions of that vector? Some $t$ dimensions? It is a bit unclear to me.

### Choosing active dimensions
I am a bit confused about how the active dimensions are being determined. What does $O(\gamma_0)$ mean in practice in Theorem 1 and 2 mean? Can't there be a huge constant with big-O notation? How do you decide that a dimension is active? I think this crucial to the entire paper, and I could not find any discussion on this.

### Qualitative experiments and visualizations
Can the authors comment on the lack of visual toy problems? It is often easier to argue with a running example for manifold learning papers. However, I believe the authors can exploit this when discussing the different active practices and how they can harm the final learned model; clear visual benefits for hypothesized techniques can be compelling.

Further, there are no qualitative experiments in the paper. It would be better if there were visual comparisons alongside the numbers. It is difficult to put the numbers by themselves in perspective without a visual yardstick.

### Synthetic Experiments
In L217, the authors say that the generated results are controlled by parameters $r, d, t,$ and $\kappa$. I am unsure how $\kappa$ influences the synthetic data generation process. Isn't $\kappa$ a parameter of the inference?

Further, I am unsure if the authors report ELBO or negative ELBO under the NLL column (L226). Also, why not report it as ELBO? If you want to claim NLL, probably do other estimation techniques like Importance Weighted ELBO or as Annealed IS-based estimator.

Also, I think the NLL values are not really informative as they cannot be compared across the rows? In each row, the data changes, and therefore there is a different target NLL value? I think a more informative experiment would be where you vary $\kappa$ and demonstrate how using more or less capacity changes the NLL, KL, and Recon for the same data?

Furthermore, if I am correct, the last block of rows should be comparable with the second block rows. So why is the reconstruction error lower when we have less number of AD than $r$?

### Real World experiments
In the introduction, L32, the authors mention how the different digits can lie on a different manifold. It would have been interesting if the authors analyzed their learned manifolds in this light.
### Writing
The writing of the introduction builds a lot of suspense about what is being addressed. While this may be nice sometimes, providing a summary of the contribution can also be beneficial without forcing the reader to go through the entire section to understand what was done and how.

The captions to tables should be more informative than they are now. For instance, in Table 3, r = 10 is implied or hidden in the text. Further, it should be mentioned which results do the tables correspond to. For example, it seems to me that Table 2 and 4 are the only real-world results and the rest are all synthetic experiments. The experimental section goes back and forth, and clear descriptions can be helpful. Another example is in Table 7. What is the true number of manifold dimensions?

# Minor things
- L32. "each could **like** on."
- L34. "expand fully in the ambient space **but become on a** low-dimensional manifold when conditioned."
- Several other typos. Please, do a thorough re-read.

---

> ### Author Response · Authors · 2022-08-02
> **Response to Reviewer R6JP**
>
> We thank the reviewer for the detailed review as well as the suggestions for improvement. Also, we appreciate that the reviewer recognize our paper as having "several interesting and important theoretical insights".
>
> Based on the review, we have uploaded a new pdf with updated definition and theorem statements. Some text descriptions are also revised. Hope this may provide a clearer presentation of our work.
>
> Next, we answer the questions in the review:
>
> - I am unclear about what it means to find an optimal solution under $\gamma = \gamma_0$ .
>
> **A**: $\gamma$ is a trainable parameter via the opimizer. We have revised Theorem 1 and 2.
>
> - What is $\varphi$ here? Is it the same as L52? Are these the first $t$ dimensions of that vector? Some $t$ dimensions? It is a bit unclear to me.
>
> **A**: It means a diffeomorphism between manifold and ambient space. Please refer to Definition 1 in the newly uploaded version.
>
>
> - How do you decide that a dimension is active?
>
> **A**: Although in numerical experiments a variance cannot be exact zero, we can still observe from a converged model that each variance of encoder is either close to 1, or close to zero. The encoder variance matrix of CVAE model on MNIST dataset is added in the new appendix, which can help to show that the active dimensions variance will approach zero when the model is converged.
>
>
> - Can the authors comment on the lack of visual toy problems?
> - Further, there are no qualitative experiments in the paper.
>
> **A**: Thanks for the suggestion. Due to bandwidth and space limitation we didn't include visualization in the paper in submission and author respond period. We'll improve the paper with visualizations in its finalized version.
>
>
> - In L217, the authors say that the generated results are controlled by parameters $r, d, t$ and $\kappa$. I am unsure how  influences the synthetic data generation process. Isn't $\kappa$ a parameter of the inference?
>
> **A**: Thanks for pointing it out. The data generation process is determined by $r, d$ and $t$. $\kappa$ is a hyper-parameter we set in the experiments. It is a typo and we have revised it.
>
>
> - Further, I am unsure if the authors report ELBO or negative ELBO under the NLL column (L226).
>
> **A**: In the updated version, we have replace `NLL` with `nagetive ELBO` and hope it is clearer.
>
>
> - Also, I think the NLL values are not really informative as they cannot be compared across the rows?
>
> **A**: Yes, we agree that comparing ELBOs under different manifolds can be problematic. However, here we reported negative ELBO as a conventional reported figure, which is not the central part of our analysis and is not for comparison.
>
>
> - Furthermore, if I am correct, the last block of rows should be comparable with the second block rows. So why is the reconstruction error lower when we have less number of AD than $r$?
>
> **A**: In Table 1, the reconstruction error in the second block is lower than that of the last block, i.e. the reconstruction error is actually higher when we have less number of AD than $r$.
>
>
> - In the introduction, L32, the authors mention how the different digits can lie on a different manifold. It would have been interesting if the authors analyzed their learned manifolds in this light.
>
> **A**: Thanks for the suggestion. Our experiment indeed shows that digits may lie on different manifolds. We will improve the paper with relevant analysis in its finalized version.
>
>
>
> - The writing of the introduction builds a lot of suspense about what is being addressed.
>
> **A**: Thanks for the suggestion.  We will improve the paper with updated introduction in its finalized version.
>
> - The captions to tables should be more informative than they are now.
>
>
> **A**: Thanks for pointing out the missing in Table 7. We have added the data information of each experiments in the captions.
>
>
> **Minor things**
> - L32. "each could like on."
> - L34. "expand fully in the ambient space but become on a low-dimensional manifold when conditioned."
> - Several other typos. Please, do a thorough re-read.
>
> **A**: Thanks for catching them, and we have fixed them in the updated version.

---

> > ### Comment · Reviewer_R6JP · 2022-08-08
> > **Thank you for the response.**
> >
> > I have updated my score to reflect the other reviews and the author response.

---

> > > ### Author Response · Authors · 2022-08-09
> > > **Reply to Reviewer R6JP**
> > >
> > > Thanks for your continued engagement with our paper. And per the reviewer's suggestion, we can certainly update the paper to include the discussed changes, noting that NeurIPS allows for an additional (10th) page to address reviewer comments in the final version.

---

### Meta-Review · Area_Chair_9Mj4 · 2022-08-31

**Recommendation:** Accept
**Confidence:** Less certain

**Metareview:**

**Summary**: This paper studies the behavior of variational auto-encoders (VAEs) and conditional VAEs (CVAEs) when trained on data that embeds a low-dimensional manifold into a higher-dimensional space. The authors demonstrate that VAEs are able to learn the intrinsic manifold dimension of the data at optimality. They also show that a similar result exists for CVAEs, and that effective conditioning should reduce the loss function at optimality. The paper then examines some common design choices in VAEs and CVAEs, observing that conditioned and unconditioned priors are theoretically equivalent, learning the decoder variance should result in better performance, and that a common weight-sharing technique in autoregressive models should be avoided. The experiments section directly addresses their claims, particularly on the synthetic dataset with known ground truth intrinsic manifold dimension.

**Strengths**: Reviewers were in agreement that this is fairly original work that yields several interesting and important insights about the ability of VAEs to estimate the intrinsic dimension of a data manifold [R6JP,ZAHe,pxuR]. Reviewer [ZAHe] sees Theorem 1 as a major contribution; it demonstrates that well-optimized VAEs can estimate the intrinsic manifold dimension, under the assumption that we have a reliable method to estimate the number of active latent dimensions. The reviewer also notes taht Theorem 2 part (i) is sensible and intuitive; adding good conditioning variables can reduce the loss, and that Theorem 3 is also an interesting result that appears to shed light on an issue with a common practical technique. Moreover, reviewer [ZAHe] notes that empirical results are strong and that the introduction and motivations are very clear, and the overall structure is easy to follow.

**Weaknesses**: The main criticisms from reviewers focused on numerous issues with clarifty, with many examples given by each reviewer [R6JP,pxuR]. Reviewer [R6JP] notes that  the paper would be easier to follow if it included some form visualization on a toy problem and included some qualitative experiments, and that there are several aspects of writing that could be improved. The introduction takes too long to explain what the contribution is. Captions could be more informative (examples given). Reviewer [ZAHe] finds that while theoretical results link intrinsic dimension of data to activations in the latent space, there is no corresponding result that links activations in the latent space to the dimensionality of generated data. This is true in both Theorem 1 and in Theorem 2 / Corrolary 2.1. This reviewer also notes a large number of minor issues and/or addressable weakenesses (19 examples given) .


**Author Reviewer Discussion**: The authors provided clarifications on many points to reviewer [R6JP] and have updated the manuscript accordingly. In response to reviewer [ZAHe] they clarified how dimesionality of generator manifold follows from latent activations, fixed the error in corrolary 2.1 pointed out by reviewer, and provided numerous responses to other questions and comments. Reviewer [pxUR] comments that while the contribution relative to (Dai & Wipf ICML 2019) regarding active dimensions is a little  oversold, the paper makes other contributions.

Reviewer [R6JP] updated their score 5->6. After extnsive discussion, reviewers [ZAHe] and [pxuR] also updated their scores 5->6 and 4->5.

**Reviewer AC Discussion**: Reviewers were in consensus that author responses had improved the paper. All reviewers indicate that consider this paper above the bar for acceptance, but do think that this paper is somewhat borderline and could also be rejected.

**Overall Recommendation**: The AC is of the opinion that the evaluation and discussion that has taken place for this paper is sufficiently thorough, and will follow the recommendations by reviewers. This is a paper that is just about above the bar for acceptance, but may also need to be rejected to make room for other papers.

**Award:**

No

---

### Decision · Program_Chairs · 2022-09-14

Accept